# *PatientProfiler:* building patient-specific signaling models from proteogenomic data

Veronica Lombardi [1], Lorenzo Di Rocco [2,8], Eleonora Meo [3,8], Veronica Venafra[1,3,4,8], Elena Di Nisio [1], Valerio Perticaroli[1], Mihail Lorentz Nicolaeasa[3,4], Chiara Cencioni[5], Francesco Spallotta [1,6], Rodolfo Negri[1,7], Francesca Sacco [3] & Livia Perfetto [1]✉

## Abstract

Deciphering patient-specific mechanisms of cancer cell reprogramming remains a crucial challenge in systems oncology, as it is key to improving patient diagnosis and treatment. For this reason, comprehensive and patient-specific multi-omic characterization of tumor specimens has become increasingly common in clinical practice. Here, we developed *PatientProfiler*, a computational workflow that integrates proteogenomic data with curated causal interaction networks to generate mechanistic models of signal transduction for individual patients. *PatientProfiler* allows multi-omic data analysis and standardization, generation of patient-specific mechanistic models of signal transduction, and extraction of network-based prognostic biomarkers. We successfully benchmarked the tool on proteogenomic and clinical data derived from 122 biopsies of treatment-naïve breast cancer, available through the CPTAC portal. We identified patient-specific mechanistic models that recapitulate oncogenic signaling pathways. In-depth topological exploration of these networks revealed seven subgroups of patients, associated with unique transcriptomic signatures and distinct prognostic values. We identified well-known Basal-like 1 and 2 subtypes, while also highlighting distinct mechanistic drivers such as the MYC–CDK4/6 axis or NF-kappaB-mediated inflammatory programs. Beyond breast cancer, *Patient-Profiler* offers a generalizable framework to transform cohort-level multi-omic data into interpretable mechanistic models, making it applicable across diverse cancer types and other complex diseases.

**Keywords** Signal Transduction; Multi-omic Integration; Mechanistic Modeling; Prognostic Biomarkers; Breast Cancer
**Subject Category** Computational Biology

## Introduction

The aim of personalized and precision medicine is to shape treatments over molecular markers of a disease at single patient resolution (Mathur and Sutton, 2017).Traditionally, precision oncology relies on histological, genomic, and single transcript biomarkers due to their scalability and cost-effectiveness. However, these approaches often result in generalized stratification and a subset of patients not benefiting from patient-drug matching (Kornauth et al, 2022). Several observations suggest that cancer onset and progression cannot be solely explained by genetic alterations but involve a complex reprogramming of signaling pathways (Sever and Brugge, 2015). Indeed, patients with similar pathway alterations are likely to converge on the same clinical outcome (Neophytou et al, 2021; Meric-Bernstam et al, 2012). These findings underscore the significance of mechanistic models and pathway analysis to systematically address the signaling events dysregulated in individual patients, to identify more granular biomarkers, and achieve the milestone of personalized therapeutic solutions (Zheng et al, 2022).

In recent years, several independent groups (Hyeon et al, 2022; Jayavelu et al, 2022) as well as the Clinical Proteomic Tumor Analysis Consortium (CPTAC) invested in a comprehensive and multi-omic (e.g., genomic, transcriptomic, phosphoproteomic, etc.) profiling of cancer specimens (Li et al, 2023), offering the unique and unprecedented opportunity of systematically and in an unbiased way assessing how cellular pathways are perturbed in a given malignancy, in general, and in individual patients.

The increasing availability of this type of data inspired the development of a range of bioinformatics tools capable of pinpointing potential prognostic biomarkers by applying clustering strategies (Ikotun et al, 2023; Hotelling, 1933; van der Maaten and Hinton, 2008; Argelaguet et al, 2018; Murtagh and Contreras, 2017). Alternative strategies integrate multi-omic layers to generate hypotheses about signaling mechanisms in the form of networks (Brooks-Warburton et al, 2022; Pai et al, 2019; Dugourd et al, 2021; Montagud et al, 2022; Drake et al, 2016). The main hypothesis behind these approaches is that to understand carcinogenesis or

[1]Department of Biology and Biotechnologies 'Charles Darwin', Sapienza University of Rome, Laboratory affiliated to Istituto Pasteur Italia-Fondazione Cenci Bolognetti, 00185 Rome, Italy. [2]Department of Statistical Sciences, Sapienza University of Rome, 00185 Rome, Italy. [3]Department of Biology, University of Rome "Tor Vergata", Rome, Italy. [4]Ph.D. Program in Cellular and Molecular Biology, Department of Biology, University of Rome 'Tor Vergata', Rome, Italy. [5]Institute of System Analysis and Informatics "Antonio Ruberti", National Research Council (IASI-CNR), 00185 Rome, Italy. [6]Istituto Pasteur Italia-Fondazione Cenci Bolognetti, Sapienza University, 00185 Rome, Italy. [7]Institute of Molecular Biology and Pathology (IBPM), National Research Council (CNR) of Italy, 00185 Rome, Italy. [8]These authors contributed equally: Lorenzo Di Rocco, Eleonora Meo, Veronica Venafra.✉E-mail: livia.perfetto@uniroma1.it

cancer progression, we can use mechanistic models. These models aim to describe experimental data via networks of regulatory interactions among proteins of the biological system under study. They function as connectors to link genotypes to phenotypes at cellular, tissue, or organism levels. Recently we developed *SignalingProfiler* 2.0 (Venafra et al, 2024), a computational workflow that combines genomic, phosphoproteomic, and transcriptomic data with networks of causal interactions derived from SIGNOR resource (Lo Surdo et al, 2022) to extract context-specific models of signal transduction.

As a matter of fact, the identification of patient-specific deregulated signaling mechanisms, and their translation into granular biomarkers, remains an evolving challenge that hampers the design of personalized therapeutic strategies (Eduati et al, 2020).

Here we present *PatientProfiler*, a computational workflow that can: (1) handle and harmonize patient-specific data; (2) apply *SignalingProfiler* 2.0 to extract patient-specific protein activities and generate mechanistic networks integrating cancer phenotypic hallmarks (Appendix Fig. S1); (3) apply clustering studies to define stratification groups and (4) deliver novel and signaling-driven transcriptomic biomarker signatures.

As a use case, we successfully applied *PatientProfiler* to a discovery cohort of 122 patients affected by breast cancer, one of the most commonly diagnosed cancers among women (Krug et al, 2020). Very briefly, breast tumors are traditionally classified into Luminal A-like tumors (LumA-l), Luminal B-like tumors (LumB-l), HER2-like tumors (HER2-l), and Basal-like (Basal-l), defined by immunohistochemistry or mRNA levels of key receptors (ESR1, PGR and ERBB2) and proliferative biomarkers (Sørlie et al, 2001). Overall, this classification plays a vital role in considering prognosis, treatment selection, and clinical trial design and has proven to be successful in many patients (Nolan et al, 2023). However, this classification remains insufficient in a large fraction of patients, and recent studies highlighted the need for further stratification especially within the Basal-l subtype, which lack standardized treatment regimens. Notably, in 2011 Lehmann and collaborators (Lehmann et al, 2011) demonstrated that triple-negative breast cancer (TNBC), which largely overlaps with Basal-like group, forms at least six distinct molecular subtypes, including two Basal-like subgroups (i.e., BL1 and BL2) with different molecular dysregulations and therapeutic characteristics (Lehmann et al, 2016, 2021; Wang et al, 2019).

Using *PatientProfiler*, we systematically captured molecular differences across the classical breast cancer subtypes. We generated one mechanistic signaling model for each patient in the cohort and employed a network-based strategy to stratify Basal-like patients into the BL1 and BL2 subgroups, thereby refining classification and opening new opportunities for more personalized therapeutic approaches.

# Results

## *PatientProfiler*, workflow overview

*PatientProfiler* is a novel R workflow designed to systematically integrate cancer-derived multi-omic data at the patient-resolution level. The scope of *PatientProfiler* is to get mechanistic insight into pathways and molecular mechanisms that are deregulated in individual patients, giving as output network-based prognostic biomarkers, and, possibly, formulation of novel therapeutic strategies in oncology.

The entire workflow is freely accessible and available for reuse and interoperability at https://github.com/SaccoPerfettoLab/PatientProfiler.

Here, we provide a step-by-step description of the method, which consists of five key steps, organized into two main parts. The first part aims to generate patient-specific models from multi-omic data (Fig. 1, Steps 1–3). Whereas the second part leverages the so-generated models to deliver biomarkers of the disease (Fig. 1, Steps 4–5).

### Step 1: Harmonization of input data

The main bottleneck toward the application of bioinformatics tools derives from the heterogeneity of available data, which often is sparse, outdated, inconsistent, and distributed in different formats. The first module of *PatientProfiler* offers optional functions that ensure and facilitate standardization of input transcriptomic, and (phospho)proteomic data.

Due to the lack of multiple replicates and of matched normal samples in some datasets, we used sample-wise z-scores (centered on the median of the expression of all the genes across the same sample/patient) to estimate up- or down-regulated analytes. Briefly, this represents a previously validated strategy that highlights patient-specific alterations while reducing technical variability (Krug et al, 2020). Importantly, this approach enables meaningful comparisons across independent cohorts despite the absence of a shared reference. To date, *PatientProfiler* can take as input multi-dimensional data (multiple data modalities, multiple patients, multiple analytes), yet analyzing each patient individually.

Users can select as input a custom dataset as well as choose among a number of pre-processed cancer data. At the time of writing *PatientProfiler* grants access to pre-harmonized proteogenomic data from more than 1600 patient-derived samples, organized in 10 different tumor types, derived from CPTAC data, for access and analysis of future users (Li et al, 2023) (Fig. 1A).

Importantly, although this step was optimized to access cancer data, we generalized it to allow broader applications, not limited to oncology cases.

### Step 2: Protein activity inference

As a second step, *PatientProfiler* systematically derives the activity of proteins from the integration of pre-processed data with prior knowledge information deposited in public repositories. More in detail, *PatientProfiler* leverages the footprint-based and the PhosphoScore approaches implemented in our recently developed *SignalingProfiler* 2.0 (Venafra et al, 2024), to estimate the activity of kinases, phosphatases and other phosphorylated proteins from phosphoproteomic datasets; and the activity of transcription factors from transcriptomic datasets (Fig. 1B) (a more detailed description of *SignalingProfiler* 2.0 is provided in Appendix Fig. S1).

The result of this step is a list of patient-specific signaling proteins modulated in their activity in the diseased tissue (Fig. 1C). Importantly, the workflow is highly flexible and users might opt to use alternative and complementary methods to estimate protein activity (Badia-i-Mompel et al, 2022).

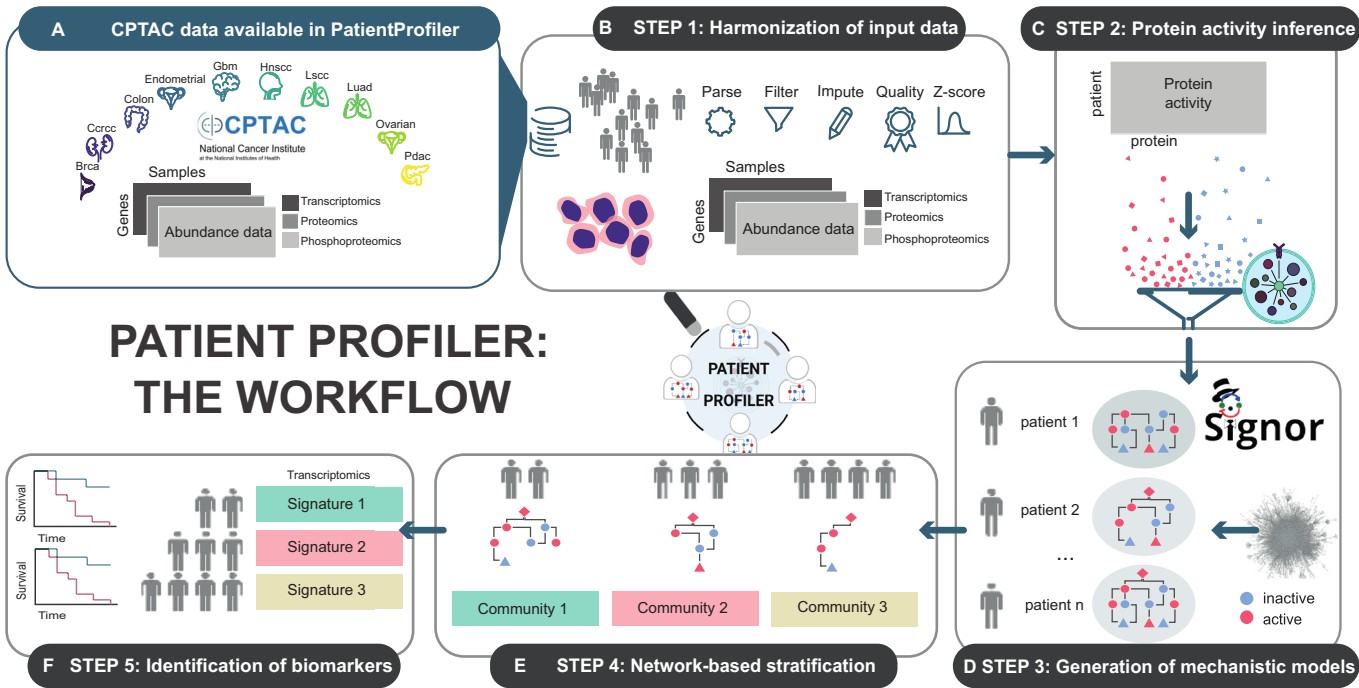

**Figure 1. PatientProfiler, the workflow.**

(A) Summary of CPTAC data (Li et al, 2023) available in *PatientProfiler*. (B) STEP 1 allows users to parse custom data or data imported from CPTAC. Data accepted: transcriptomics, proteomics, and phosphoproteomics from multiple samples. (C) In STEP 2 *PatientProfiler* allows the user to estimate, within each sample, protein activities from the multi-omic data (red: active proteins; blue: inactive proteins). (D) In STEP3 *PatientProfiler* leverages causal data in public resources (Lo Surdo et al, 2022; Keshishian et al, 2021) to hierarchically connect protein activities from STEP 2 and to build for each sample (patient) a mechanistic model. (E) In STEP 4 *PatientProfiler* leverages mechanistic models from STEP 3 to detect communities based on network structural similarities. Each community is represented by a group of samples (patients) and by a subgraph capturing key signaling interactions within the group. (F) In STEP 5, communities detected in STEP 4 are used in combination with relative gene expression profiles to identify biomarker transcriptomic signatures. Source data are available online for this figure.

### Step 3: Generation of mechanistic models

As a further step, *PatientProfiler* leverages *SignalingProfiler* 2.0 (i) to retrieve, from the SIGNOR repository, causal interactions that can explain the experimentally observed changes in protein activities; and (ii) to estimate the activation level of hallmark phenotypes (e.g., Apoptosis, Proliferation). These interactions are then hierarchically organized to derive a collection of coherent networks portraying the molecular mechanisms underlying patient-specific disease development (Fig. 1D).

So generated networks can be directly visualized in Cytoscape, in two formats: (i) the first displays the complete generated network; (ii) the second highlights circuits that lead to the specific deregulation of user-selected phenotypes.

Importantly, thanks to the usage of visual features (e.g., color and style of the nodes and edges), these networks facilitate the interpretation, while reducing complexity and noise, of the input data.

### Step 4: Network-based stratification

The essence of the second part of *PatientProfiler* (Fig. 1, Steps 4–5) is to identify network-based features that can explain differences among groups of patients, thus providing prognostic biomarkers. The premise behind *PatientProfiler* is that patients displaying similar deregulated pathways are likely to display similar clinical outcomes and should be treated similarly. *PatientProfiler* offers the

opportunity to run agnostic clustering analysis, to identify novel stratification groups (communities) associated with common network modules.

Specifically, *PatientProfiler* builds bipartite graph representing patient-to-interaction relationships and applies the Louvain algorithm. The Louvain algorithm offers a scalable solution for partitioning large graphs into communities by maximizing modularity, without requiring the number of clusters to be specified in advance. Importantly, each resulting community is defined by a group of patient-nodes and the corresponding interaction-nodes, which together represent a signaling module consistently shared by that specific subset of patients (Fig. 1E).

### Step 5: Identification of biomarkers

In this last step, *PatientProfiler* offers the user the chance to derive network-driven transcriptomic signatures to use as biomarkers (Fig. 1F). The identification of biomarker transcripts is particularly important since transcriptomic datasets are more common in clinical practice, where at the current moment only a very limited number of patients can be profiled at the proteogenomic level. Here, *PatientProfiler* exploits an ANOVA followed by a post hoc Tukey test to identify transcripts (i.e., a transcriptomic signature) whose overexpression is associated with each of the communities derived in Step 4.

# Breast cancer use case, the dataset

**Figure 2. Breast cancer use case, STEPS 1–2.**

(A) Cartoon representing the scope of the use case. (B) Breast cancer subtypes: traditional subtyping include Luminal A-like tumors (LumA-l)(ESR1+, PGR+ and ERBB2-, low proliferative index); Luminal B-like tumors (LumB-l)(ESR1+, PGR+/- and ERBB2-, high proliferative index); HER2-like tumors (HER2-l)(ESR1-, PR- and ERBB2+); and Triple-negative breast cancer or Basal-like (Basal-l)(ESR1-, PGR- and ERBB2-, high proliferative index) whose features include high invasiveness, high metastatic potential, proneness to relapse, and poor prognosis. (C) Breast cancer data was retrieved from *PatientProfiler* or from the CPTAC portal using the python "cptac" package (Li et al, 2023). (D) Omic level size, relative to transcriptomic, proteomic, and phosphoproteomic data from a cohort of 122 breast cancer patients (Krug et al, 2020) (C). (E) Classification in subtypes (Basal-l, LumA-l, LumB-l, and HER2-l) and relative group size, as per original publication and referring to the NMF.Cluster. (F) Protein expression level of known biomarkers in different subtypes, after data manipulation (STEP1) (ESR1: estrogen receptor, PGR: progesterone receptor, ERBB2/HER2: human epidermal growth factor receptor 2, TOP2A: proliferative biomarker). Box plots are defined as follows: center line = median, box boundaries = interquartile range (25th–75th percentile, IQR), whiskers = minima and maxima within 1.5 × IQR from the quartiles, and dots = outliers beyond the whisker range. Statistical significance was assessed using one-way ANOVA followed by Tukey's HSD post hoc test, and BH-adjusted *p*-values are reported. Basal-l (red, *n* = 30), LumA-l (blue, *n* = 32), LumB-l (green, *n* = 37), and HER2-l (yellow, *n* = 23). (G) Estimated protein activities in individual patients, stratified by subtypes. The dashed-red bar indicates the average amount of estimated proteins in each subgroup. Red, blue, purple, and green bars refer to kinases, phosphatases, transcription factors, and other types of proteins, respectively. Subtype color code: Basal-l in red, LumA-l in blue, LumB-l in green, and HER2-l in yellow. Source data are available online for this figure.

## Application of *PatientProfiler* to breast cancer data

As a use case, we set out to analyze with *PatientProfiler* proteogenomic data derived from a discovery cohort of 122 breast cancer patients, available at the CPTAC portal (Krug et al, 2020). The scope is to demonstrate that the workflow can provide insights into the patient-specific cell reprogramming that underlies breast cancer (Fig. 2A), use this information to capture molecular differences across the intrinsic breast cancer subtypes and to stratify patients beyond this classification (Fig. 2B) and deliver novel biomarkers.

To this scope, we accessed CPTAC data available in *Patient-Profiler* (March 2023) and extracted transcriptomic, proteomic, and phosphoproteomic datasets, displaying analyte abundance of

23,121 genes, 10,107 proteins, and 38,775 phosphosites, respectively, in breast cancer patients (Fig. 2C,D). In addition, we retrieved somatic mutation and clinical information. As displayed in Fig. 2E, patients derived from the above-described cohort are homogeneously distributed across the four breast cancer subtypes: Basal-l, HER2-l, LumA-l, and LumB-l.

Most frequent mutation events in the cohort appear to be loss-of-function (LOF) and gain-of-function (GOF) mutations in the TP53 and PIK3CA genes, respectively (Appendix Fig. S2, Dataset EV2). Interestingly, while the mutation in TP53 seems to associate prevalently with Basal-l patients (Fisher's exact test, BH adjusted *p*-value = 1.963e-08) and appears significantly depleted in Luminal subtypes, the combination of the two is enriched in HER2-l patients, exclusively, and tends to anti-correlate with Basal-l

patients (Fisher's exact test, BH adjusted $p$-value = 0.02943). The remaining mutations did not show significant associations with molecular subtypes, suggesting they play only a limited role in patient stratification. This observation highlights the importance of signaling on tumorigenesis and cancer progression.

Next, we applied data pre-processing functions, implemented in *PatientProfiler* to harmonize the data, and handle missing values (Appendix Fig. S3), and, for each patient and each dataset, we computed the sample-wise z-score to infer up- or down-regulated analytes. To assess the reliability of this approach, we analyzed the z-score profiles of well-established breast cancer genes (Nolan et al, 2023). As shown in Appendix Fig. S4, the trends we observe for these genes are consistent with those generated using the TCGA dataset (Koboldt et al, 2012), supporting that our intra-sample normalization preserves meaningful biological signals across independent cohorts.

We also observed that data manipulation and normalization preserve the differences between subtypes, especially at the proteome and transcriptome levels (Appendix Figs. S4 and S5). In fact, the expression of known subgroup-associated biomarkers (ESR1, PGR, ERBB2, and TOP2A) is in line with the classification, both at the protein and transcript levels (Fig. 2B,F; Appendix Fig. S6A). ESR1 and PGR proteins appear to be significantly upregulated in LumA-l and LumB-l patients, with greater variability in the latter, ERBB2 in HER2-l, whereas Basal-l patients display lower levels of the three receptors. In line, Basal-l and LumB-l show higher levels of the proliferative TOP2A marker.

In summary, the workflow correctly maintains differences between classical subtypes and has the potential to reflect their distinct molecular profiles.

### Protein activity estimation

Analyte abundance is usually used to identify biomarkers, however, this type of data tends to be noisy and too complex to interpret (Kim et al, 2018). To reduce dimensionality and complexity, we exploited the second step of *PatientProfiler* (Fig. 1C), which leverages footprint-based and PhosphoScore analyses in *Signaling-Profiler* 2.0, to infer the activity of signaling proteins from multi-omic information (Fig. 2G). Thanks to this approach, we were able to predict the activity of 712 proteins in the entire cohort (Appendix Fig. S7, Dataset EV3). As shown, the number of predicted proteins appears quite homogeneous in the different subtypes (Fig. 2G), with LumA-l patients displaying an average increased number (170) of predicted proteins. Of note, in the four groups, most inferred proteins are kinases (mean = 98), followed by transcription factors; this might be due to a combination of factors which includes from one hand the important role of these proteins in mediating signal transduction, and, from the other hand, a regulon size bias in the prior knowledge used.

Next, we performed a one-way ANOVA to identify 280 signaling molecules (about 40% of the inferred total proteins) displaying the strongest difference in activities in different subgroups (BH correction, adjusted $p$-value < 0.05, Dataset EV3). Importantly, the test identifies known subgroup-associated biomarkers (ESR1, ERBB2, and TOP2A), showing that protein activity inference preserves subtype nature (Appendix Fig. S6B). Interestingly, cell cycle-associated proteins, such as the cyclin-dependent kinases (CDKs) and the E2F4 oncogenes appeared to be strongly dysregulated between subgroups, displaying an increased activity

in the most severe breast cancer subtype, Basal-l (Appendix Fig. S8). Similarly, MYC and MYCN oncogene transcription factors display similar behavior, suggesting stronger levels of proliferation in this subtype, in line with previous evidence (Xu et al, 2010; Schafer et al, 2020).

In line with these results, CDK2 activity appears higher in Basal-l cells (MDA-MB-231), compared to cell lines representative of other subtypes (Appendix Fig. S9A,B). This is demonstrated by enhanced phosphorylation at the activating residue pThr160 (even if not significant) and increased levels of the associated cyclin protein. Similarly, CDK5 is more active in a Basal-l cell line as revealed by increased levels of phosphorylation of the activating pTyr15 site (Appendix Fig. S9C) (Zukerberg et al, 2000).

In summary, inferred protein activities successfully reduced the complexity of the initial data, by increasing the level of abstraction and correctly representing breast cancer subtypes.

### Network generation

A key challenge in multi-omics data integration is extracting the cause-effect relationships underlying experimental data. Translated to our use case, this task attempts to address specific molecular events deregulated in individual breast cancer patients, in the form of a signed and directed graph.

To this aim, we employed Step 3 of PatientProfiler to build a collection of 122 mechanistic networks.

This collection can be accessed for analysis, visualization, and reuse via NDEx at the link https://www.ndexbio.org/#/networkset/1387170f-4c20-11f0-a218-005056ae3c32?accesskey=2dfafd3f0a022 04ef789f7b4b93741fa193013ae7de552eb60e1ea90e187302a. Importantly, for each patient in the cohort, we made available the full interactome generated and a subnetwork displaying the functional circuits impacting key cancer hallmarks (Fig. 3A; Appendix Fig. S10). In each resulting network, nodes represent protein and phenotype entities associated with an activity status and edges the regulatory interactions, at the PTM-resolution level, among them.

These models are heterogeneous in size, as displayed by the number of nodes and edges which span from 100–230 and 150–500, respectively (Fig. 3B). To note, this variability is independent of the subtype. On the other hand, the proportion of active/inactive nodes and activatory/inhibitory edges appears more homogeneous.

The so-generated networks permit easy inspection of patient-specific circuits linking mutated proteins to dysregulated proteins and eventually to altered cancer hallmarks. Thus, to determine whether the so-generated models could recapitulate the malignant behavior of the tumor samples as well as the different severity of the four subtypes, we set out to compare the activation levels of a number of cancer phenotypes (Hanahan and Weinberg, 2011). As shown in Fig. 3C, patients-specific models have the capability to recapitulate a malignant context, as revealed by increased levels of pro-oncogenic and pro-invasive phenotypes, namely proliferation and epithelial-mesenchymal transition, along with the suppression of apoptosis and cell-cycle exit. Also, in line with expectations, Basal-l models (red bars in Fig. 3C) are associated with more aggressive behavior for all considered phenotypes.

In summary, *PatientProfiler* provides a framework portraying biologically meaningful molecular mechanisms underlying patient-specific disease development. Networks facilitate the readability and interpretability of multi-omic data while reducing complexity and

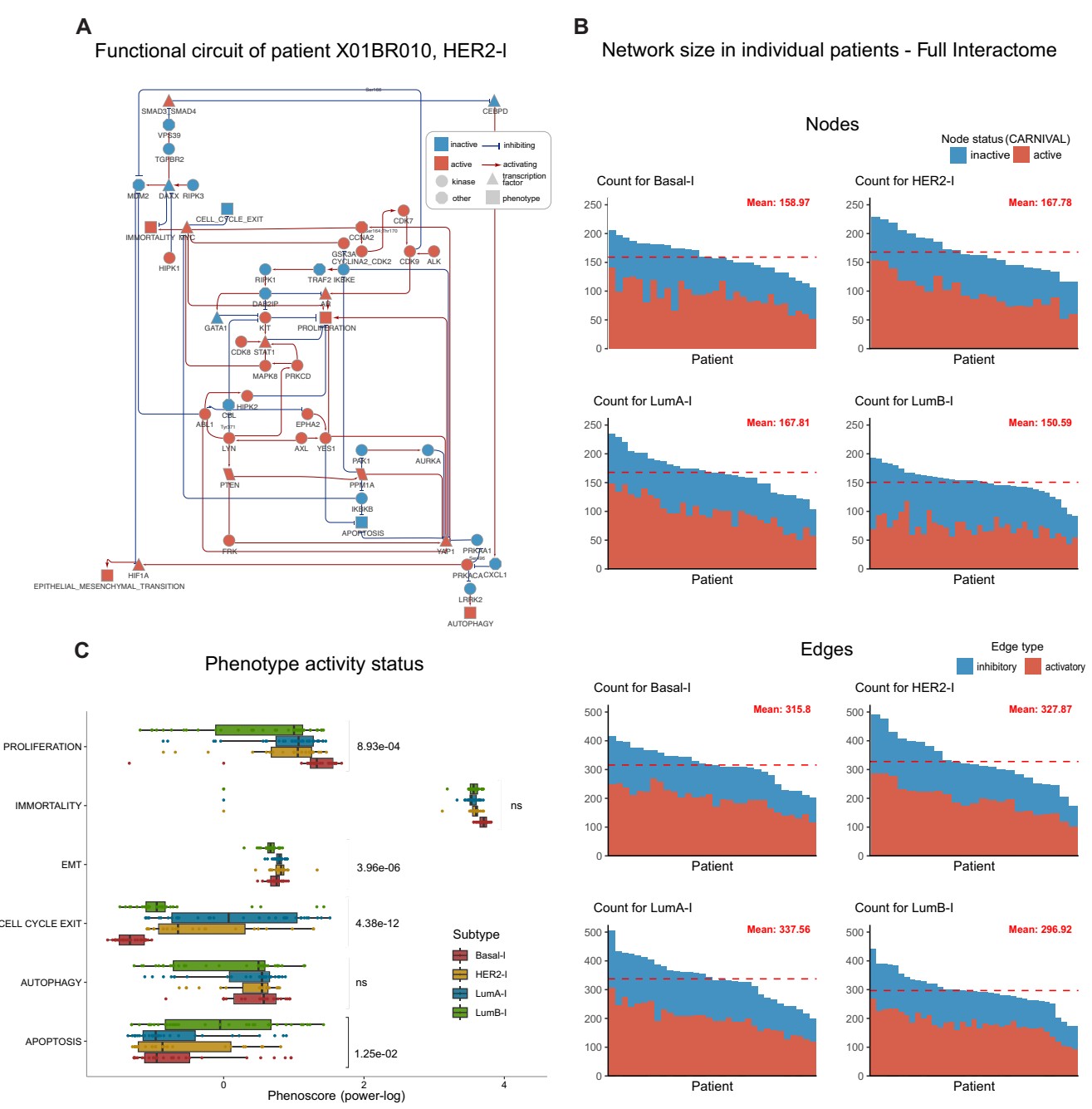

**Figure 3. Breast cancer use case, mechanistic models' generation (STEP 3).**

(A) Example of a mechanistic model generated with *PatientProfiler*. The graph depicts the functional circuits to key phenotypes (rectangles) in the HER2-I patient X01BR010. Red and blue nodes represent active and inactive proteins, respectively; red arrows and t-shaped edges represent activatory and inhibitory relationships, respectively. Detected phosphorylated residues are shown as edge labels (e.g., Ser668). (B) Summary of the network size of the full interactomes, as shown by the distribution of the number of nodes (top) and edges (bottom) in individual patients, stratified by subtypes. The dashed-red bar indicates the average number of estimated nodes and edges in each subgroup. (C) Activation level (power-log of the PhenoScore) of key hallmark phenotypes in different subtypes as measured in the STEP3 of the *PatientProfiler* workflow. Box plots are defined as follows: center line = median, box boundaries = interquartile range (25th–75th percentile, IQR), whiskers = minima and maxima within 1.5 × IQR from the quartiles, and dots = outliers beyond the whisker range. Statistical significance was assessed using one-way ANOVA and BH-adjusted *p*-values are reported. Sample size (*n*) of Basal-I (red), HER2-I (yellow), LumA-I (blue) and LumB-I (green) for each category: Proliferation (*n*: 30, 23, 32 and 37); Immortality (*n*: 30, 23, 32 and 37); EMT (*n*: 29, 23, 32 and 37); Cell cycle exit (*n*: 30, 23, 32 and 37); Autophagy (*n*: 23, 23, 31 and 36); Apoptosis (*n*: 29, 21, 29 and 33). Source data are available online for this figure.

noise and offer novel features that can be used to determine similarities and differences between and within the four breast cancer subtypes.

### Communities' prioritization and transcriptome signatures extraction

One of personalized medicine goals is the identification of more granular biomarkers capable of stratifying patients at diagnosis in a more specific way, to better predict prognosis and therapeutic interventions (Savchenko and Bunimovich-Mendrazitsky, 2024; Willis and Lord, 2015). This is especially true in the case of breast cancer, where at diagnosis patients are assigned one subtype based on immunohistochemistry and/or expression levels, with no complete success in predicting disease outcome and drug response.

Since patients displaying similar deregulated pathways are likely to show a similar evolution of the disease as well as a similar response to therapies (Alyass, 2015), we established a community detection analysis to identify novel breast cancer subgroups that might reveal network-derived prognostic biomarkers.

Leveraging the previously generated collection of regulatory networks, we stratified the input cohort into distinct communities. Specifically, we employed the community detection approach (STEP 4 of *PatientProfiler*) to stratify the 122 patients of the discovery cohort according to the topological structure of their models (Fig. 4A). To ensure meaningful community structures, we empirically optimized the resolution parameter of the Louvain algorithm, selecting the value that maximized the modularity of the resulting partition (Appendix Fig. S11).

By this approach, we were able to identify seven communities of patients who are sharing common network topological structures (Fig. 4B; Appendix Fig. S12, Dataset EV4, NDEx link: https://www.ndexbio.org/#/networkset/dd734d60-4c24-11f0-a218-005056ae3c32?accesskey=573cb8bc7f0977857ac19a6a6b65c231320fe8baced6a756aaf32570e0a8fa18). These communities appear heterogeneous and include different subtypes, even if each of them is characterized by the significant overrepresentation of one subtype over the others (Fig. 4B; Appendix Fig. S14). We tested if communities were associated with clinical and molecular covariates. Community 2 (showed significant association with the Tumor Mutational Burden (BH adjusted $p$-value = 0.013). No other covariates (Stage, Age, Ethnicity) were significantly associated with any communities.

Since our clustering strategy is based on shared interactions rather than shared genes, the same gene may appear in multiple communities, but always within distinct regulatory contexts. As shown in Appendix Fig. S13, this occurrence is limited. Indeed, the percentage of shared genes across communities is low, suggesting a distinct molecular identity for each group.

Furthermore, since our workflow models the active and inhibited forms of each gene separately, the same gene can appear in both states within a single community, depending on the regulatory interactions it is involved in across different patients. We refer to these as ambiguous nodes. As shown in Appendix Fig. S12, the overall frequency of ambiguous nodes remains low across most communities and it tends to increase with network size, possibly reflecting a degree of variability or noise in the networks.

To determine a biological difference between the detected communities, we inspected the level of phenotype activations in these subgroups, and we observed that Community 2 and Community 4 are associated with increased levels of pro-oncogenic and pro-invasive phenotypes (Fig. 4C), indicating that they might represent two subgroups of patients with the worst disease outcome. Interestingly, these two communities are enriched in Basal-l patients, highlighting the heterogeneity of Basal-l patients in their signaling rewiring and the necessity to identify biomarkers to a better stratification.

To this aim, we used the so-obtained communities to extract relative transcriptomic signatures and benchmark them over an independent dataset (validation cohort). Briefly, for each community, we considered the transcriptomic profile of the community members and, by performing an ANOVA followed by a Tukey post hoc test, we identified transcripts whose up-regulation displayed the highest variance when compared to the remaining patients in the cohort.

By this approach, we were able to identify seven community-derived signatures (Appendix Fig. S15, Dataset EV4).

To investigate the unique pathways characterizing Signatures 2 and 4 and their relative communities (representing the two subgroups of Basal-l patients), we performed (i) an over-representation analysis of the identified signatures (Appendix Fig. S16), (ii) a comparison of deregulated cancer hallmarks (Fig. 4C), and (iii) an examination of circuits driving pro-oncogenic phenotypes, at the network level (Fig. 5A; Appendix Fig. S17). These analyses converge on two critical findings: in Community 2, there is a pronounced activation of cell growth and cell-cycle progression mediated by the MYC-CDK4/CDK6 axis, while in Community 4, a robust pro-inflammatory response driven by the NF-kappaB pathway. At the network level, we observed activation of the NF-kappaB pathway downstream of ERK1/2 and TBK1. Transcriptomic analysis revealed upregulation of macrophage activation markers, genes such as S100A8, S100A2, KRT81, KRT6C, MARCO and MMP1, as well as processes related to keratinization and the formation of a cornified envelope in breast cancer cells. Collectively, these findings suggest that Community 4 is associated with a metaplastic phenotype, characterized by increased aggressiveness and invasiveness (Zhang et al, 2012).

### Signature on an independent Breast Cancer cohort

Next, to demonstrate a broader application of our findings, we exploited the so-obtained signature to classify patients in an independent dataset. Briefly, we accessed transcriptomic, clinical, and follow-up data from The Cancer Genome Atlas (TCGA) (Koboldt et al, 2012), relative to a validation cohort of 1094 breast cancer patients. As described in the method section, we first used the signature to stratify the TCGA cohort in relative subgroups (Appendix Fig. S18, Dataset EV4). Next, by leveraging follow-up data we performed a survival analysis to detect whether the different signatures-derived stratification groups are associated with a different clinical outcome. As displayed in Fig. 5B, the Kaplan–Meier curve indicates a strong association of Signature2 and Signature4 with a worse overall survival, with Signature2 associated with a better prognosis. Remarkably, TCGA patients enriched by Signature2 and Signature4 are both associated with Basal-l subtype classification (Appendix Fig. S18). Furthermore, by applying Fisher's exact test, we showed that TCGA patients enriched for Signature2 and Signature4 were significantly associated with the basal-like BL1 (BH correction, adjusted $p$-value = 1.22e-39) and BL2 (adjusted $p$-value = 1.38e-11) subtypes,

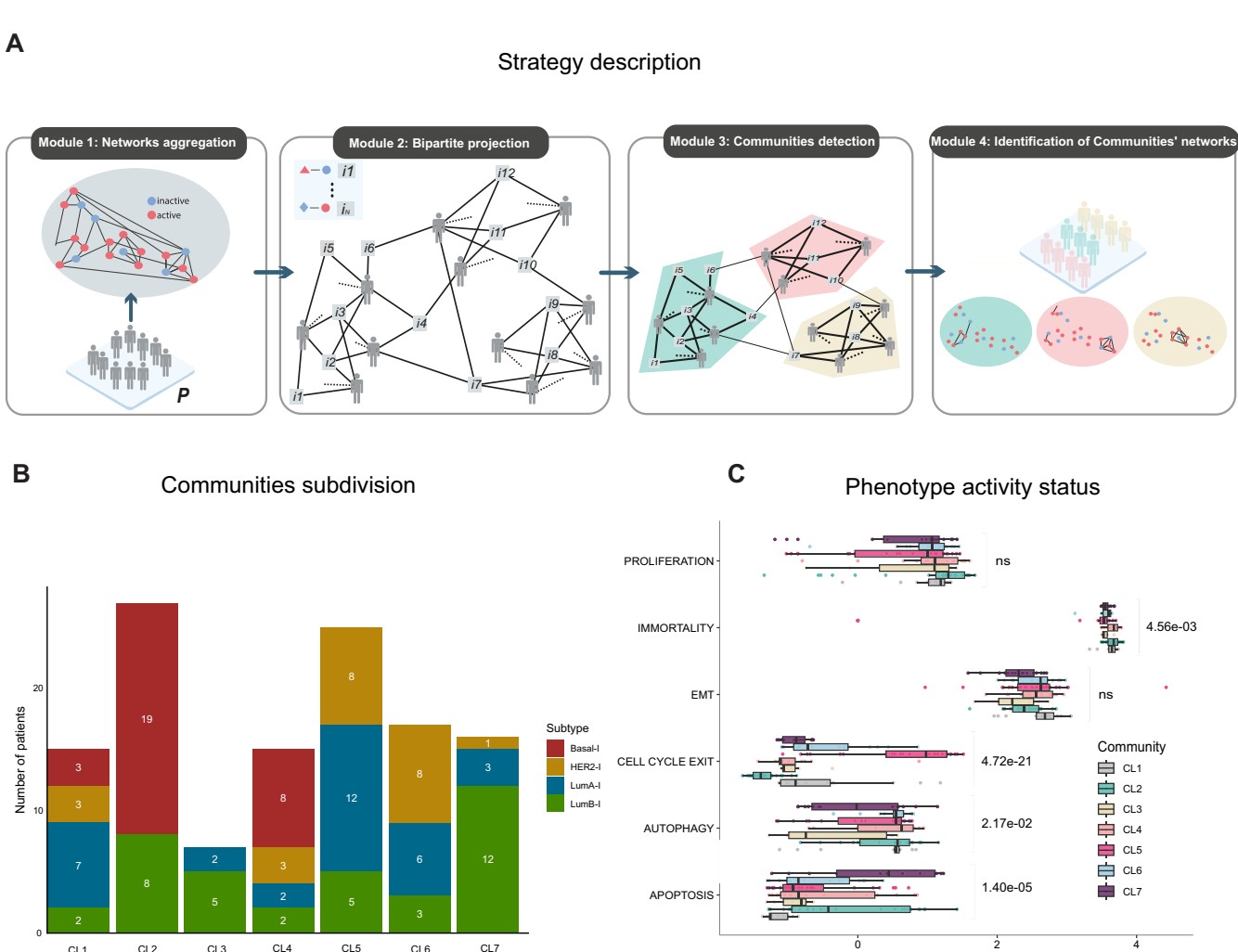

**Figure 4. Community detection and their clinical impact (STEP 4).**

(A) Cartoon depicting the community detection algorithm implemented in *PatientProfiler*. Interactions from input patients are aggregated into a single network and projected into a bipartite patient-interaction graph, where each edge connects a patient-node to an interaction-node included in their specific regulatory network. Finally, modularity optimization is applied to identify distinct patient communities and their corresponding sub-interactomes. (B) Our cohort of 122 Breast Cancer patients is divided into seven communities (CL1-7). Bar plots depicting the subtype composition of each community. Subtype color code refers to the NMF. Cluster classification, as derived from the original publication (Krug et al, 2020): Basal-I in red, LumA-I in blue, LumB-I in green, and HER2-I in yellow. (C) Activation level (power-log of the PhenoScore) of key hallmark phenotypes in different communities as derived from the mechanistic models. Box plots are defined as follows: center line = median, box boundaries = interquartile range (25th–75th percentile, IQR), whiskers = minima and maxima within 1.5 × IQR from the quartiles, and dots = outliers beyond the whisker range. Statistical significance was assessed using one-way ANOVA and BH-adjusted *p*-values are reported. Relative sample size (*n*) of Community 1 (CL1, gray), CL2 (green), CL3 (yellow), CL4 (pink), CL5 (magenta), CL6 (blue) and CL7 (purple) for each category: Proliferation (*n*: 15, 27, 7, 15, 25, 17 and 16); Immortality (*n*: 15, 27, 7, 15, 25, 17 and 16); EMT (*n*: 15, 26, 7, 15, 25, 17 and 16); Cell cycle exit (*n*: 15, 27, 7, 15, 25, 17 and 16); Autophagy (*n*: 15, 20, 7, 15, 23, 17 and 16); Apoptosis (*n*: 13, 26, 7, 14, 24, 15 and 13). Source data are available online for this figure.

respectively, proposed in previous studies (Wang et al, 2019; Lehmann et al, 2011). This suggests that the network-based stratification can identify more granular subgroups of Breast cancer patients within the broader Basal-like classification.

Importantly, *PatientProfiler* associates each subgroup with a signaling map that illustrates specific pathway alterations (Fig. 5A), thus improving the interpretability and translational value of the findings. To confirm the link between the signatures and the relative maps, we established to analyze reverse phase protein array data (RPPA) derived from the very same TCGA patients classified according to Signature2 and Signature4 and determine their

coherence with the node state in the CL2 and CL3 networks. Importantly, only 12 out of 23 and 5 out of 13 of the CL2 and CL4 nodes, respectively, had been analyzed in the RPPA dataset. We next computed the Student's t-test to compare their abundance in the two groups. In line with the maps, and as shown by (Lehmann et al, 2021) CyclinE and MYC appear up-regulated in Signature2-enriched patients, whereas ERK2/MAPK1, PKCaplha/PRKCA and EGFR display an up-regulation in Signature4-classified patients (Fig. 5C).

Altogether, our results suggest that the network-based stratification approach implemented via *PatientProfiler* can identify

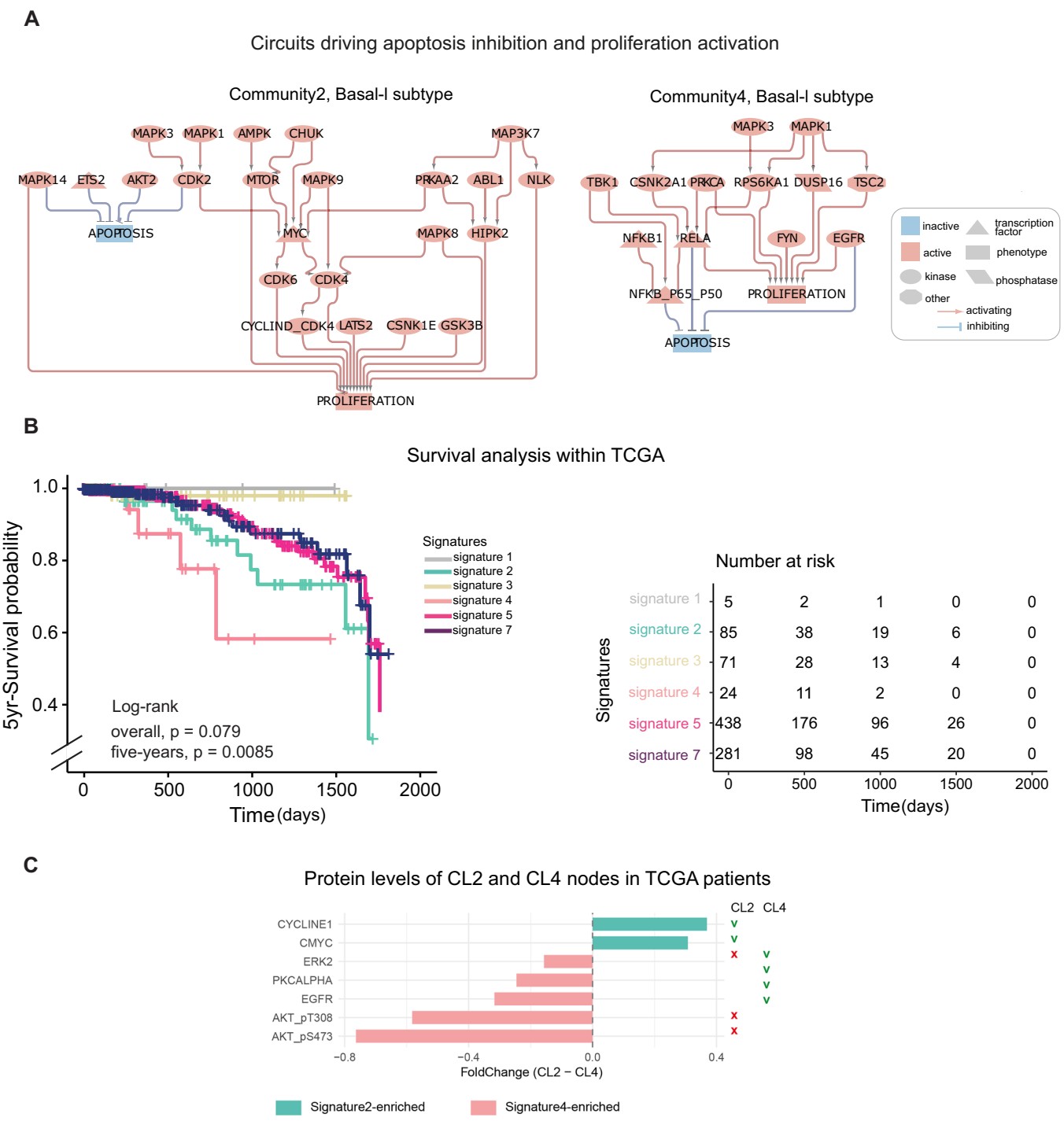

**Figure 5. Circuits driving oncogenic phenotypes and Signature application on a validation Breast Cancer cohort.**

(**A**) Mechanistic models generated with *PatientProfiler*. The graph depicts the functional circuits to key phenotypes (rectangles) in Community 2 and Community 4. Red and blue nodes represent active and inactive proteins, respectively; red arrows and t-shaped edges represent activatory and inhibitory relationships, respectively. (**B**) Kaplan–Meier plot showing survival probability of Breast Cancer patients from The Cancer Genome Atlas (TCGA) (Koboldt et al, 2012) stratified in subgroups by Signatures1–7. (**C**) Difference in protein levels of CL2 and CL4 nodes extracted from the map in (**A**), as derived from the Reverse Phase Protein Arrays (RPPA) of TCGA patients (Koboldt et al, 2012). Proteins up-regulated in Signature2- and Signature4-enriched TCGA patients are displayed in light blue and pink, respectively. On the right, the symbology indicates whether the protein level is coherent (green v) or incoherent (red x) with the activity in the map in (**A**). Only significant results (BH correction, adjusted *p*-value < 0.05) are displayed. Source data are available online for this figure.

## Randomization results

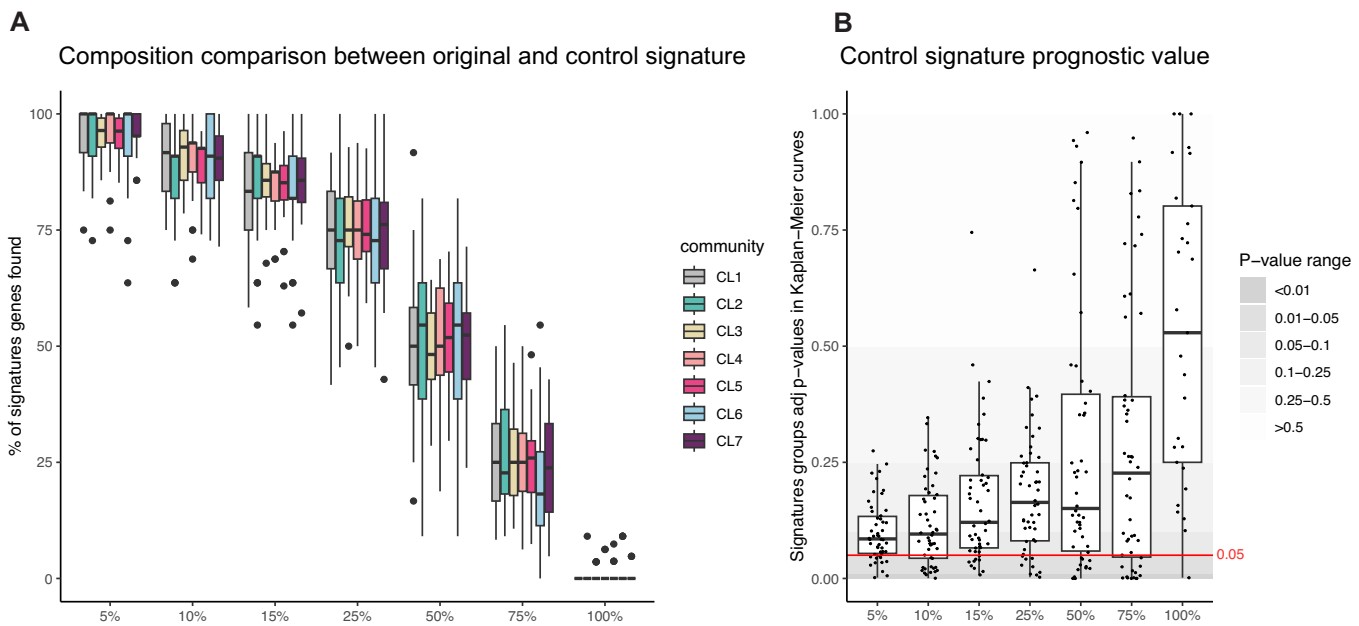

**Figure 6. Randomization analysis result.**

(A) Percentage of genes from the original *PatientProfiler* run recovered in control signatures across 50 randomization runs at increasing proportion of shuffled genes (5, 10, 15, 25, 50, 75, 100%) in CPTAC transcriptomics data. Box plots are defined as follows: center line = median, box boundaries = interquartile range (25th–75th percentile, IQR), whiskers = minima and maxima within 1.5 × IQR from the quartiles, and dots = outliers beyond the whisker range. Relative sample size (*n*) of Community 1 (CL1), CL2, CL3, CL4, CL5, CL6 and CL7 for each level of randomization: 5% (*n*: 50, 50, 50, 50, 50, 50 and 50); 10% (*n*: 50, 50, 50, 50, 50, 50 and 50); 15% (*n*: 50, 50, 50, 50, 50, 50 and 50); 25% (*n*: 50, 50, 50, 50, 50, 50 and 50); 50% (*n*: 50, 50, 50, 50, 50, 50 and 50); 75% (*n*: 50, 50, 50, 49, 50, 50 and 50); 100% (*n*: 49, 20, 50, 39, 42, 49 and 34). (B) Prognostic value of control signatures evaluated via Kaplan–Meier analysis. Each boxplot summarizes the adjusted *p*-values across runs for different shuffling percentages: 5% (*n*: 50, min *p*-value: 1.85e-3, max *p*-value: 0.275), 10% (*n*: 50, min *p*-value: 4.77e-4, max *p*-value: 0.346), 15% (*n*: 50, min *p*-value: 7.64e-3, max *p*-value: 0.745), 25% (*n*: 50, min *p*-value: 2.65e-3, max *p*-value: 0.664), 50% (*n*: 50, min *p*-value: 3.62e-5, max *p*-value: 0.960), 75% (*n*: 50, min *p*-value: 5.11e-11, max *p*-value: 0.948), 100% (*n*: 29, min *p*-value: 1.50e-3, max *p*-value: 1). Box plots are defined as follows: center line = median, box boundaries = interquartile range (25th–75th percentile, IQR), whiskers = minima and maxima within 1.5 × IQR from the quartiles, and dots = outliers beyond the whisker range. The horizontal red dashed line marks the 0.05 significance threshold. Background shading indicates *p*-value ranges. Statistical significance was assessed using log-rank test and BH adjusted *p*-values are reported. Source data are available online for this figure.

clinically and molecularly distinct subgroups of breast cancer patients. These subgroups not only correlate with established molecular classifications such as BL1 and BL2, but also provide deeper mechanistic insights through pathway-specific alterations supported by proteomic data.

### Robustness assessment

To assess the robustness of *PatientProfiler* at each analytical step, we evaluated the sensitivity to noise through randomization and cross-validation analyses. At the protein inference and network construction steps (Step 2 and 3), we previously demonstrated that *SignalingProfiler* is highly sensitive to randomization of both experimental input data and prior knowledge networks, resulting in significant reduction in precision and recall (Venafra et al, 2024). Building on this, we further evaluated robustness at the community and signature generation steps (Steps 4 and 5). Increasing proportions of randomized gene-expression data (5, 10, 15, 25, 50, 75, 100%) led to progressively smaller overlaps between randomized control signatures and the original signatures (Fig. 6A), accompanied by reduced prognostic value (Fig. 6B). As a support analysis and to assess the robustness of the community-

specific signatures, we performed a cross-validation procedure based on random sampling. For each *PatientProfiler* community, we randomly split the CPTAC patients into training (70%) and test (30%) sets (50 times). For each split, we derived community-specific signatures using the training set and used these signatures to classify test patients. We then evaluated: (i) signature robustness, whether the signatures generated from the 70% training set were similar to those obtained from the full dataset; (ii) consistency in patient assignment, whether the test patients were assigned to the same communities as in the original full run. The scope of this analysis is to assess the sensitivity of the signatures to patient subsampling, highlighting both their strengths and limitations in reliably stratifying patients across different cohorts. Cross-validation analysis further revealed that most patient communities maintained moderate to high stability, with good recovery of key signature genes and F1 score > 0.5 across 50 runs (Appendix Fig. S19A,B). Performance tended to improve with larger community size and balanced signature complexity (Appendix Fig. S19C,D). Community 6, however, showed poor gene and patient recovery and was not identified in TCGA cohort (Appendix Fig. S18), suggesting lower stability and cohort specificity.

All together, these analyses indicate that biologically and clinically meaningful signals are sensitive to increasing levels of noise, and cross-validation confirms that the signatures and patient assignments are overall stable and reproducible.

## Comparison with other methods

In the previous paragraphs, we provided evidence that a deep understanding of patient-specific signaling dynamics is crucial to achieve mechanistic insights into disease mechanisms and to pinpoint clinically relevant biomarkers. Traditionally, novel stratification groups and molecular signatures have been identified by data-driven techniques, such as unsupervised clustering (e.g., KMeans or the hierarchical clustering) (Ikotun et al, 2023; Murtagh and Contreras, 2017), alone or coupled with dimensionality reduction techniques like the MOFA, PCA, or t-SNE (Hotelling, 1933; van der Maaten and Hinton, 2008; Argelaguet et al, 2018). Among them, MOFA has proven particularly effective in identifying associations between molecular features and clinical phenotypes. Despite being powerful from the prognostic perspective, these methods are not conceived to formulate novel mechanistic hypotheses.

More recently, a new class of approaches has emerged that aims at generating sample-specific networks and/or models to address translational challenges while digging into the molecular mechanisms of diseases. This second group of methods includes *PatientProfiler*, together with PROFILE, netDx, iSNP, pCHIPS and COSMOS (Brooks-Warburton et al, 2022; Pai et al, 2019; Dugourd et al, 2021; Montagud et al, 2022; Drake et al, 2016). Briefly, they combine prior-knowledge interaction data with genomics, transcriptomics, and phosphoproteomics to create contextualized networks or models. In Appendix Fig. S20, we summarize crucial differences between *PatientProfiler* and related tools. The development and proliferation of these tools underscore a growing need to uncover dysregulation at the sample-specific level. In particular, PROFILE uses personalized Boolean models to simulate the effects of drug interventions on patient-specific cancer phenotypes. netDx is a machine learning method that integrates multi-omics patient data to build a patient classifier that predicts clinical features (e.g., low vs high risk assessment). iSNP maps the functional impact of single nucleotide polymorphisms (SNP) and identifies areas of a global regulatory network enriched by the patient-specific ensemble of variants to derive a network footprint to identify patient clusters with distinct pathomechanisms. pCHIPS enables protein activity estimation and visual inspection and prioritization of the signaling pathways specific to each individual patient and is useful for suggesting personalized treatment options. COSMOS connects data-driven analysis of multi-omic data with systematic integration of mechanistic prior knowledge.

To note, tools such as netDx, pCHIPS and COSMOS require integration with other approaches to derive stratification groups, whereas *PatientProfiler* and iSNP go further and leverage the collection of generated networks to identify signaling-driven communities of patients and to propose biomarkers.

All together, these methods vary in the approach, in the purpose and in the type of input data they accept. It is important to highlight that, as previously outlined, some tools were not developed for biomarker detection and rather for different and complementary purposes. The comparison indicates that

*PatientProfiler* appears as a valuable method to simultaneously generate patient-specific directed networks with therapeutic prioritization potential and can generate transcriptomic signature biomarkers.

# Discussion

As cancer is a signaling disease, patients with similar pathway alterations are likely to display similar evolution of disease and clinical outcomes. Deciphering patient-specific mechanisms of cancer cell reprogramming is, therefore, pivotal in oncology to improve diagnosis and treatment. In this paper, we thoroughly present *PatientProfiler*, a computational workflow that leverages causal interaction data to address how the genetic and molecular background of individual patients contributes to the establishment of a malignant phenotype. *PatientProfiler* is an open-source, R-based package composed of several functions that allow for multi-omic data analysis and standardization, generation of patient-specific mechanistic models of signal transduction, and extraction of network-based prognostic biomarkers.

Here, we show that *PatientProfiler* is a modular workflow that allows users to analyze custom multi-omics and multi-factorial data as well as to choose among pre-processed proteogenomic data from more than one thousand patient-derived samples, organized in ten different tumor types, as derived from CPTAC data. It is important to emphasize that even though *PatientProfiler* was optimized for cancer data analysis, it is generally applicable and might be exploited in broader contexts.

In the first module, the *PatientProfiler* enables the estimation of protein activity as well as the generation of personalized models portraying the molecular mechanisms underlying patient-specific disease development.

We applied *PatientProfiler* to proteogenomic profiles of Breast cancer patients, available at CPTAC, and produced a collection of 122 publicly available mechanistic networks. This collection represents, per se, a goldmine of information that can be used to further explore mechanisms of tumorigenesis and cancer progression. This collection, by displaying protein nodes according to their activity status and according to their hierarchical role in impacting cancer hallmark phenotype, might support the prioritization of personalized therapies. Remarkably, we demonstrated that phenotypes are biologically relevant and coherently reflect disease severity. Moreover, when the emerging features can be recapitulated on stable cancer cell lines, as is the case for the differential CDKs activity, this can endorse experimental work to gain additional mechanistic details and perform preliminary drug screening.

In the second module, *PatientProfiler* leverages the generated collection of regulatory networks to stratify the input cohort into distinct communities, each associated with a common subgraph recapitulating the dysregulated pathways and with a biomarker transcriptomic signature that can be used to classify novel patients. We applied this second module to our collection of 122 personalized models and detected seven communities of Breast cancer patients. Our community-based stratification, derived from mechanistic multi-omic modeling, partially overlaps with intrinsic subtypes (as shown by enrichment analysis) and captures orthogonal features. This discrepancy might be due to the limited

capabilities of *PatientProfiler* (discussed below), but it could also mirror the inadequacy of the current classification in capturing the complexity and the heterogeneity of Breast Cancer patients and might reflect the added resolution provided by our integrative approach.

These communities pinpointed two subgroups of Basal-l patients (Community 2 and 4), enriched by relative transcriptomic signatures (Signatures2 and 4), and characterized by distinct prognostic values. Moreover, we showed that TCGA patients enriched by Signature2 and Signature4 significantly associate with well-known Basal-l subgroups, namely BL1 and BL2 (Lehmann et al, 2011; Wang et al, 2019). By this analysis, we demonstrated that *PatientProfiler* enables the systematic characterization of such subgroups while providing mechanistic insight into the signaling events regulated in each subgroup and in each individual, with possible impact for therapy formulation. Specifically, in Community 2 we observe sustained cell growth and cell-cycle progression mediated by the MYC-CDK4/CDK6 axis, while in Community 4, the activation of a NF-kappaB-mediated pro-inflammatory response and of pathways driving a metaplastic phenotype, with a resulting increase in aggressiveness and invasiveness. Importantly, these observations are in line with results from Lehmann et al (2021) and the activation of these axes was also partially confirmed by looking at the RPPA profiles of the patients.

Moreover, by implementing a randomization strategy and a cross-validation analysis, we showed that *PatientProfiler* exhibits strong robustness across multiple analytical steps, particularly in community and signature generation. The sensitivity analyses demonstrate that signature composition and their clinical relevance are sensitive to increasing degrees of input noise, and cross-validation confirms that the signatures and patient assignments are generally stable and reproducible.

We extensively described how *PatientProfiler* can be useful to help researchers and clinicians to extract actionable insights from their data. Despite that, it is important to stress that the workflow presents limitations. The first constraint is represented by the prior knowledge space. Indeed, *PatientProfiler* suffers from the restricted coverage of available regulon and interaction information in public resources. As a matter of fact, presently *PatientProfiler* has the potential to include approximately 7500 protein nodes (<50% of the UniprotKB proteome) and 43,145 potential regulatory interactions in a network. This is a strong underestimation with respect to the estimated number of molecular interactions occurring in cells (Bier et al, 2015). To alleviate this burden, we implemented regular updates to ensure prior knowledge data is maintained up to date.

An additional important point to note is that *PatientProfiler* adopts sample-wise z-score normalization as a proxy of analytes' fold-change. This choice was mainly guided by the absence, in many datasets, of multiple replicates and matched control samples. Importantly, this approach has been adopted in similar contexts (Krug et al, 2020) and enables the construction of patient-specific molecular profiles while mitigating technical variability and batch effects across samples. A crucial limitation of this strategy is that the results are relative to the internal variance of each sample, which may reduce the power and the reliability in comparing expression levels to a baseline condition or across patients from different cohorts. This should be considered when inferring protein's activity, since sample-centric normalization may obscure global shifts relevant to network-level interpretation.

Another important limitation to mention is that, so far, *PatientProfiler* only considers transcriptomics and (phospho)proteomics, thereby overlooking the regulatory mechanisms described in the models. This aspect critically impacts the capability of *PatientProfiler* to detect effective communities and, consequently, transcriptomic signatures. As shown, some signatures did not enrich patients in independent datasets. Additional -omic levels such as epigenetics, metabolomics, and ubiquitylomics (among others) are becoming more common and popular (Papadaki et al, 2025) and should be considered for future developments of the workflow.

Furthermore, our current community detection strategy relies on the Louvain algorithm to identify groups of patients. While effective, this approach is primarily based on shared interactions between patients. Alternative clustering strategies, that go beyond direct edge comparison by capturing the global topology and higher-order patterns of each network, are being considered for future developments. As a final remark, we aim to stress that given the flexibility of the *PatientProfiler*, users have the possibility to apply different clustering methods. As an example, we aim to compute compact embeddings of patient-specific mechanistic networks into a continuous space. By measuring distances in this embedding space, we can assess similarities between patients at the network level more comprehensively, ultimately enhancing the tool's predictive performance.

To conclude, *PatientProfiler* enables clinicians and researchers to stratify patients according to key signaling events and demonstrates that patients with similar pathway alterations are associated with different prognoses. *PatientProfiler* addresses the emergent need to extract interpretable networks and derive biologically relevant information from complex multi-omics data and meets the challenge of generating "one model for one patient", posing the basis for future development in the personalized medicine field.

# Methods

**Reagents and tools table**

| Reagent/resource | Reference or source | Identifier or catalog number |
|---|---|---|
| **Antibodies** | | |
| Rabbit anti-CDK5 | Cell Signaling | 2506 |
| Rabbit anti-Phospho-CDK5 (Tyr15) | Invitrogen | PA577909 |
| Rabbit anti-Phospho-CDK2 (Thr160) | Cell Signaling | 2561 |
| Mouse anti-CDK2 | SantaCruz | sc-6248 |
| Mouse anti-cyclin A | SantaCruz | sc-271682 |
| Goat anti-Rabbit IgG (H + L) Secondary Antibody, HRP | Invitrogen | 31460 |
| Goat anti-Mouse IgG (H + L) Secondary Antibody, HRP | Invitrogen | 31430 |
| **Cell cultures** | | |
| MCF7 (*H. sapiens*) | | N/A |
| MDA-MB-231 (*H. sapiens*) | | N/A |

| Reagent/resource | Reference or source | Identifier or catalog number |
|---|---|---|
| MDA-MB-361 (*H. sapiens*) | | N/A |
| Mycoplasma PCR Detection Kit | Abm | G238 |
| Dimethyl Sulfoxide for cell culture | PanReac AppliChem | A3672,0250 |
| DMEM, 1X (Dulbecco's modification of Eagle's medium) with 4.5 g/L glucose & sodium pyruvate without L-glutamine, 500 mL | Corning | 15-013-CV |
| RPMI 1640, 1X without L-glutamine, 500 mL | Corning | 15-040-CV |
| 100 mL Trypsin EDTA 1X | Corning | 25-053-CI |
| Dulbecco's PBS (1x), - Ca/Mg - Phenol Red, 500 mL | Sial | SIAL-PBS-1A |
| L-Glutamine solution | Corning | G7513-100ml |
| Fetal Bovine Serum (FBS) SUPERIOR, 500 ML SOUTH AMERICA | Sial | YOURSIAL-FBS-SA |
| 100 mm TC-treated Culture Dish | Corning | 430167 |
| **Chemicals and other reagents** | | |
| PageRuler Prestained Protein Ladder | Pierce | 26616 |
| 2-Mercaptoethanol | Fluka | 63689 |
| 4X Laemmli Sample Buffer | Bio-Rad | 1610747 |
| Albumin (BSA) Fraction V (pH 7.0) for Western blotting | PanReac AppliChem | A6588,0100 |
| Skim Milk Powder | Sigma-Aldrich | 70166-500G |
| Sodium azide | Sigma-Aldrich | S2002-5G |
| Acrylamide/Bis Solution, 29:1 (30% w/v, 3.3% C) | Serva | 10687.01 |
| UltraPure TEMED | Invitrogen | 15524-010 |
| Resolving Gel Buffer | Bio-Rad | 161-0798 |
| Stacking Gel Buffer | Bio-Rad | 161-0799 |
| 10% SDS Solution | Cell Signaling | 20533S |
| Ammonium Persulfate | ThermoFisher Scientific | 17874 |
| Nitrocellulose membrane, Roll, 0.45 µm, 30 cm×3.5 m | Bio-Rad | 1620115 |
| Clarity Western ECL Substrate 500 mL | Bio-Rad | 170-5061 |
| Tween 20 | Bio-Rad | 1706531 |
| Cell Lifter | Corning | 3008 |
| Pierce RIPA Buffer | ThermoFisher Scientific | 89901 |
| cOmplete Mini, EDTA-free Protease Inhibitor Cocktail Tablets | Roche | 11836170001 |
| PhosSTOP EASYpack | Roche | 4906837001 |
| β-Glycerophosphate disodium salt hydrate | Sigma-Aldrich | G9422-10G |
| Ponceau S solution | PanReac AppliChem | A2935,0500 |

| Reagent/resource | Reference or source | Identifier or catalog number |
|---|---|---|
| Bradford Protein Assay Reagent | Pierce | 1856209 |
| 10X Tris/Glycine/SDS Buffer | Bio-Rad | 1610772 |
| 10X Tris/Glycine Buffer | Bio-Rad | 1610771 |
| Methanol (Reag. USP, Ph. Eur.) for analysis, ACS, ISO, BioChemica | Applichem | 131091 |
| **Software** | | |
| Image Lab 6.1 | https://www.bio-rad.com/it-it/product/image-lab-software?ID=KRE6P5E8Z | N/A |
| GraphPad Prism 7.0c | https://www.graphpad.com/ | N/A |
| Office Microsoft Excel | https://www.microsoft.com/it-it/ | N/A |
| Cytoscape 3.10.3 | https://cytoscape.org/ | N/A |
| R 4.4.3 | https://www.r-project.org/ | N/A |
| Python 3.7 | https://www.python.org/ | N/A |
| **Other** | | |
| MicroCL 17R Microcentrifuge | ThermoFisher Scientific | 75002456 |
| ChemiDoc MP Imaging System | Bio-Rad | 17001402 |

## Workflow description

We introduce *PatientProfiler*, an R-based workflow composed of two main parts: The first part aims to generate patient-specific models from multi-omic data (Steps 1–3). Whereas the second part leverages the so-generated models to deliver biomarkers of the disease (Steps 4–5).

The code, the documentation and a tutorial are available at: https://github.com/SaccoPerfettoLab/PatientProfiler.

All the functions along with the explanation of each associated parameter and default values have been outlined in Dataset EV1.

### Step 1: Dataset manipulation

Here, *PatientProfiler* harmonizes omic data through the "omics_update" function, which (i) parses the data to make it compliant with further steps in the workflow; (ii) filters and handles missing data; (iii) performs quality checks and (iv) performs z-scoring to estimate deregulated analytes.

More specifically, for transcriptomic data it is possible to apply (i) parsing (ii) filtering and (iii) z-score computation. For proteomics and phosphoproteomics (i) parsing (ii) quality check (iii) imputation, and (iv) z-score computation. For phosphoproteomics users can also retrieve the sequence window.

**Handling missing data**. For transcriptomics, genes exceeding a threshold of missing values (default = 80%) (Appendix Fig. S3) are removed to exclude analytes consistently absent in most of the patients, remaining missing values are replaced with 0. For proteomics and phosphoproteomics missing values are imputed using the "mice" package with the default method set to "pmm"

(predictive mean matching) (van Buuren and Groothuis-Oudshoorn 2011).

**Quality checks.** Here, the "omics_update" function performs quality checks on annotated sequences in phosphoproteomics and on identifiers and symbols used. For all omics, the function updates gene symbols and retrieves the corresponding UniprotKB identifier if not already provided (uniprot_idx, default = NULL). For phosphoproteomics it performs additional steps to ensure consistency: with the default parameters: "pep_col_name" = NULL to specify the Peptide sequence name and "sw_len" = 7 to define the length of the sequence window (15-mer centered on the phosphorylated amino acid), the function updates aminoacid and position of the phosphorylation and, if the column Peptide is present, retrieves the sequence window according to the latest UNIPROT sequence. More specifically, the steps include: (i) remove phosphosites with unclear phosphorylated ammino acid; (ii) update gene names to primary symbols and add UniprotKB identifiers and full protein sequences using UniprotKB API queries; (iii) for multiple peptide sequences, retain the longest; (iv) for duplicated entries, retained phosphosite with lower multiplicity; (v) generate the sequence window from the full protein sequence; (vi) update the ammino acid sequence and position based on the defined sequence window.

**Z-score calculation.** Here the "omics_update" function performs z-scoring to estimate deregulated analytes. It's possible to perform z-scoring via the "zscore" parameter (default = TRUE) and choose whether to center using the median or mean ("metric", default = "median", as more robust). Z-scoring can be performed by columns (sample-wise, i.e., intra-patient) or by rows (gene-wise, i.e., across transcripts/proteins/phosphosites) according to the parameter "metric". Values greater or less than $+/-1.96$ (i.e., $+/-2$ standard deviations) are considered significantly modulated, as they fall outside the 95% confidence interval ($p$-value < 0.05). For instance, in a sample-wise z-score, this allows detection of genes/proteins/phosphosites whose abundance significantly deviates from the sample-specific distribution, potentially indicating dysregulation within that sample.

### Steps 2–3: Protein activity inference and generation of mechanistic models

The second and third steps of *PatientProfiler* exploited *SignalingProfiler* 2.0R package (Venafra et al, 2024) to generate patient-specific signed and oriented graphs connecting molecular entities (e.g., proteins, complexes, metabolites) and ending up on functional traits (phenotypes).

In step 2, we developed the wrapper function for the *SignalingProfiler* 2.0R package (Venafra et al, 2024) "extract_protein_activity", which infers patient-specific protein activities modulations from multi-omics data combining footprint-based methods and PhosphoScore algorithms.

The function takes as inputs highly processed omic datasets derived from step 1 and returns for each patient the list of activated and inhibited proteins.

The function "exract_cohort_activity" then extracts the protein activity for each patient and from each omic level combining footprint-based analysis: (i) TFEA and KSEA to infer the activity of transcription factors, kinases and phosphatases from regulons in the transcriptomics and phosphoproteomic datasets through VIPER inference. Briefly, VIPER (Virtual Inference of Protein-activity by Enriched Regulon analysis), is a statistical framework developed by Alvarez et al (2016) for the accurate assessment of

protein activity from gene expression data; (ii) PhosphoScore calculation, to predict the activity of other phosphorylated proteins by the modulation of regulatory phosphosites in the phosphoproteomic dataset (Appendix Fig. S1).

In particular, "extract_protein_activity" handles multi-patients, multi-omics datasets. The list of associated parameters and default values have been outlined in Dataset EV1.

In this phase it is possible to define parameters for the TFEA, KSEA and PhosphoScoore analyses.

TFEA parameters ("tf_params") include:

- "organism" for reference species (default = "human");
- "reg_minsize" for minimum regulons group size (default is 10 to ensure robust results);
- "exp_sign" whether to consider or not only significant results in the transcriptomics (default = FALSE to consider the entire distribution, as per VIPER);
- "collectri" indicates whether to collect source regulon (TF-TG relationships) from the collecTRI resource (Müller-Dott et al, 2023) (default = FALSE, which uses Dorothea and SIGNOR datasets) (Garcia-Alonso et al, 2019; Lo Surdo et al, 2023);
- "hypergeom_corr" whether to correct results with an hypergeometric test, to reduce the impact of the regulon size (default = TRUE, as it was shown to produce more accurate results);
- "GO_annotation" whether to include GO annotations (default = TRUE, to derive protein functions from GO:MF) (Gene Ontology Consortium et al, 2023);
- "correct_proteomics" whether to correct with proteomic data (default =FALSE, this parameter refers to a method that gives less importance to phosphosites equally modulated in proteomics and phosphoproteomics, it applies only to KSEA).

KSEA parameters ("kin_params") include:

- "organism" for reference species (default = "human");
- "reg_minsize" for minimum regulons group size (default is 5 to ensure robust results);
- "exp_sign" whether to consider or not only significant results in the proteomics (default = FALSE to consider the entire distribution, as per VIPER);
- "integrated_regulons" indicates whether to collect source regulon (kinase-phosphosites relationships) from the KINOME atlas project (Johnson et al, 2023; Yaron-Barir et al, 2024) (default = TRUE, to increase regulon coverage);
- "hypergeom_corr" whether to correct results with an hypergeometric test, to reduce the impact of the regulon size (default = TRUE, as it was shown to produce more accurate results);
- "GO_annotation" whether to include GO annotations (default = TRUE, to derive protein functions from GO:MF);
- "correct_proteomics" whether to correct with proteomic data (default =FALSE, to avoid misuse in TFEA, this parameter refers to a method that gives less importance to phosphosites equally modulated in proteomics and phosphoproteomics, it applies only to KSEA, where it is recommended to set TRUE).

PhosphoScore parameters ("phosphoscore_params") include:

- "organism" for reference species (default = "human");
- "activatory" whether to consider only phosphosites that impact

on protein activity and to exclude phosphosite that impact on protein stability (default = TRUE, to ensure consistency with TFEA, where protein activity is considered);
- "GO_annotation" whether to include GO annotations (default = TRUE, to derive protein functions from GO:MF);
- "correct_proteomics" whether to correct with proteomic data (default =FALSE, to avoid misuse in TFEA, this parameter refers to a method that gives less importance to phosphosites equally modulated in proteomics and phosphoproteomics, it applies only to KSEA, where it is recommended to set TRUE).

The result of this step 2 is a file for each patient containing the gene_name, the Uniprot ID, the molecular functions (mf), the "final_score" (a score for the inferred activity) and method that describes how the activity score is calculated.

In step 3 we developed the wrapper function "create_cohort_-networks" which builds patient-specific mechanistic models connecting each patient's functionally annotated mutations to the inferred signaling proteins.

The "create_cohort_networks" takes as input:

- "sources": a user-defined set of perturbed molecules/receptors (e.g., targets of a treatment or mutated genes in a disease);
- "activities": the dataset of proteins whose activity has been estimated in Step 2 or measured by alternative tools;
- "desired_phenotypes": the list of phenotypes to connect, as available in the SIGNOR resource (default = NULL);
- "transcriptomics", "proteomics" and "phosphoproteomics": the dataset to filter paths based on the available experimental data (default = NULL).
- a number of parameters that will be used in nested function (as detailed below).

The function "create_cohort_networks" internally calls five ad hoc nested functions:

a) "get_PKN" function is needed to generate the PKN, the prior-knowledge space onto which *PatientProfiler* will build the mechanistic model. Here the definition of the relative parameters (to be specified at the user level as a list within the "PKN_options" parameter:
  - "preprocess" whether to remove interactions involving entities not detected in at least one omic layer (default = TRUE, to ensure context-dependent models);
  - "direct" whether to retain only direct interactions, as described in the SIGNOR resource (default = TRUE, to give relevance to signal transduction relationships; to detect transcriptional regulations is recommended to set this parameter to FALSE);
  - "organism" (default = human);
  - "with_atlas" whether to integrate Ser/Thr and Tyr atlas inferred interactions (default = FALSE to limit a bias for phosphorylation reactions (Johnson et al, 2023; Yaron-Barir et al, 2024);
  - "custom" whether to provide a custom PKN (default = FALSE, by default *PatientProfiler* uses the SIGNOR interactome).
b) "create_naive_network" is a nested function needed to render the PKN context-specific, retaining only interactions that are responsible for the modulation of transcription factors, kinases, phosphatases, and other proteins (as derived from Step 2). Users have the possibility to embed in the network a set of starting

perturbed nodes. The result of this function is a collection of patient-specific naive networks. Here the definition of the relative parameters (to be specified at the user level as a list within the "naive_options" parameter):
  - "layers" To specify the number of layers from one to three: (1) one layer of connection, usually connects user-defined sources to user-defined targets, (2) two layers of connections, usually retrieve interactions between sources and an intermediate level and between the intermediate level and targets; (3) three layers of connections, usually connects first sources to targets using two intermediate levels (default = 2, as discussed in Venafra et al, 2024);
  - "max_lenght" Vector of length = layers, number of steps allowed in each layer (default = c(1,4), as discussed in Venafra et al, 2024);
  - "connect_all" whether connecting intermediate nodes of shortest path (default = TRUE, as discussed in Venafra et al, 2024).
c) The "optimize_network_with_carnival" function optimizes the naive networks using the CARNIVAL algorithm (Liu et al, 2019). Briefly, CARNIVAL (CAusal Reasoning for Network identification using Integer VALue programming) is a method for the optimization of regulatory signaling networks using as constraints the activation state of nodes in the network, to generate a mechanistic model. The result of this function is a collection of patient-specific mechanistic models. Here the definition of the relative parameters (to be specified at the user level as a list within the "carnival_options" parameter):
  - solver, which solver can be used between cplex, gurobi, lpSolve, cbc (Liu et al, 2019) (default = cplex);
  - carnival_type, which CARNIVAL type can be used between inverse (which optimizes the network from the user-defined targets to an unspecified source) and vanilla (which optimizes the network from the known source nodes to the target nodes) in three modalities: vanilla_one_shot (only one layer of optimization, usually sources-targets), vanilla_two_shots (two layers of optimization, it considers a user-defined intermediate level), vanilla_three_shots (three layers of optimization, it considers two user-defined intermediate levels) (default = inverse, as it requires minimum constraints).
d) The "infer_and_link_phenotypes" function leverages the Pheno-Score algorithm to infer from the model the regulation of user-defined hallmark phenotypes. Specifically, it embeds the ProxPath method (Iannuccelli et al, 2024), to measure the functional proximity of a list of gene products to target pathways and phenotypes, using causal interactions annotated in SIGNOR. The result of this function is a collection of patient-specific mechanistic models associated with dysregulated phenotypes. Here the definition of the relative parameters (to be specified at the user level as a list within the "phenoscore_options" parameter):
  - "path_length" defines the maximum number of steps to connect nodes in the model to output phenotypes (default = 3, as specified in Venafra et al (2024));
  - "stat" defines the z-score statistics, either mean or median (default = mean);
  - "zscore_threshold" defines the z-score threshold to select proximal phenotype (default = $-1.96$, two standard deviations, considered significant, as they fall outside the 95% confidence interval ($p$-value < 0.05));

- "nrandom", number of randomization (default 1000);
- "pvalue_threshold" significance threshold (default = 0.05);
- "remove_cascade" collapses paths directly linking nodes to phenotypes (default = TRUE to avoid introduction of noise in the models);
- "use_carnival_activity" whether to use final score (as derived in Step 2) or carnival activity (as derived in Step 3) to compute activation state of phenotypes (default = FALSE, to use experimentally-derived activity scores).

(e) The "format_patient_network" function that specifies formatting options for Cytoscape visualization (Shannon et al, 2003).

### Step 4: Network-based patient stratification

Here, *PatientProfiler* extracts network-based communities through the "generate_communities" function with the parameters "t_lower" as the lower threshold for filtering edges based on their frequency across patients (default = 4) and "t_upper" as the upper threshold for filtering edges based on their frequency across patients (default = 30). More in detail, this step leverages the collection of patient-specific mechanistic networks $G = \{g_i, i = 1, \cdots, N\}$ to simultaneously uncover patient clusters and extract the specific pathways highly associated with them. To this end, we design a pipeline consisting of four modules: (1) *Networks Aggregation*. In this module, we merge the collection of networks $G$ into a single aggregated network $G^*$ that stores all the interactions observed in at least one patient within the input cohort. We focus exclusively on the biological and mechanistic relevance, by excluding pathways from $G^*$ originating from an inhibited node. This approach allows us to retain only functional drivers. Removing this subset of pathways ensures the analysis focuses on functionally active relationships likely to be involved in pathological regulatory mechanisms with an actual effect on the phenotype. In addition, this refinement filters out the potential background noise hindering the identification of patient communities; (2) *Bipartite Projection*. The second module builds a bipartite graph $B = (V, E)$ to model the patient-interaction relationship. In this graph, the vertex set is defined as $V = P \cup I$, where $P$ represents the set of patient-nodes representing the members in the input cohort, and $I$ denotes the set of interactions-nodes occurring in $G^*$. In the bipartite graph, an edge $e = (p, i)$ exists between a patient-node $p \in P$ and an interaction-node $i \in I$ if and only if the corresponding patient $p$ includes the interaction $i$ in their interactome. For example, if the interaction MAP2K1 → MAPK1 (denoted as interaction $j$) is present in the regulatory model of patient $i$, the bipartite graph includes an edge between the patient-node $i$ and the interaction-node $j$. Consequently, the degree $d(j)$ of an interaction-node $j \in I$ represents the number of patients whose regulatory network contains that specific interaction. Modeling interactions as nodes in the bipartite graph allows us to connect patients based on shared molecular events, resulting in a topological structure that supports community detection driven by common mechanistic patterns. (3) *Community Detection*. The third module involves detecting communities in $B$. To achieve this, we leverage the concept of modularity (Newman and Girvan, 2004) a quality metric that evaluates the strength of a network's partition into clusters. Modularity is widely used for community detection, including in bipartite networks (Barber, 2007). By maximizing modularity in $B$, our pipeline identifies clusters of patient-nodes that are highly

interconnected through subsets of shared interaction-nodes. To compute the optimal partition, we use the Louvain algorithm (Blondel et al, 2008), which applies greedy optimization techniques to iteratively refine clusters. This hierarchical approach begins by grouping nodes into communities that maximize modularity. These communities are then aggregated into super-nodes, and the process is repeated on the resulting graph. The algorithm continues this iterative optimization until no further improvements or refinements can be achieved. To reduce noise during the clustering phase, our pipeline applies a preliminary filter to interaction-nodes in $I$, removing those with degrees below a lower threshold $t_L$ or above an upper threshold $t_U$. Indeed, interaction-nodes with low degrees could represent rare events or simply noise, lacking significance for patient stratification. Conversely, interaction-nodes with high degrees tend to be less informative, as they are common across most of the dataset. These ubiquitous interactions do not support distinguishing between patients and can hide the specific signals of the communities. Since no universally optimal thresholds exist, values must be chosen based on cohort-specific features. In our case, aiming for fine-grained patient stratification, we set $t_U$ to half the cohort size. Furthermore, we excluded interactions found in only one patient ($t_L = 1$). Concerning $t_L$, we selected the smallest threshold that resulted in a modularity score above 0.3, a commonly accepted cutoff in the literature, above which clusters are generally considered to have a meaningful modular structure. This choice ensures that we extract densely connected communities, while still retaining sufficient information and avoiding aggressive filtering that might occur with higher values of $t_L$. In our case, $t_L = 4$ resulted in a modularity value equal to 0.32, while $t_L = 2$ and $t_L = 3$ produced smaller values. Appendix Fig. S11B reports the frequency distribution of the interaction nodes, providing an estimate of those that are filtered out. (4) *Community Identification*. In the last module, we identify the specific pathways associated with each patient community by first grouping the patient-nodes assigned to the same cluster in the bipartite graph from the previous step. Next, the corresponding interaction-nodes that co-occur in the same cluster are considered as community-specific interactions. These interaction-nodes may represent isolated molecular events or combine into larger, coherent pathways that capture the underlying molecular mechanisms characteristic of each patient community.

### Step 5: Identification of biomarkers

In this last step, *PatientProfiler* offers the user the chance to derive transcriptomic signatures from the obtained communities detected in Step 4. This is implemented with the "extract_signatures" function. Specifically, this function first retrieves all the patients belonging to the community and extracts their transcriptomic profiles. Next, for each gene performs an ANOVA followed by a post hoc Tukey test and filters genes that are (i) up-regulated in the patients belonging to the community, to generate positive signatures (mean expression in the community > 0); (ii) that display the highest variance in respect to the rest of the patients (background) (Benjamini-Hochberg correction, adjusted $p$-value < 0.01); and (iii) that exceed minimum expression difference (mean expression in the community − mean expression in the background). Function parameters are:

- "padj_thres" as the $p$-value threshold for filtering significant results (default = 0.01, to be more stringent);

- "mean_exp_clus_thres" as the minimum mean expression threshold for gene filtering (default = 0, as it's assumed signatures are positively expressed);
- "max_val" as the maximum number of genes to include in each community-specific signature (default = 50).

### Network visualization

Networks were visualized in Cytoscape using the "RCy3" package (version 2.24.0) and the "createNetworkFromIgraph" function. The style "SP_pheno_layout.xml" was applied for the patient-specific mechanistic models.

## Louvain algorithm and parametrization

The Louvain algorithm detects communities by optimizing the modularity function in a greedy way. Modularity measures the difference between observed edge density within clusters and that expected under a random graph model. The general modularity function is:

$$Q = \frac{1}{2m} \sum_{i,j} \left[ A_{ij} - \gamma \frac{k_i k_j}{2m} \right] \delta(c_i, c_j)$$

where $A_{ij}$ is the adjacency matrix, $k_i$ and $k_j$ are node degrees, $m$ is the total number of edges, $\delta(c_i, c_j)$ is 1 if nodes $i$ and $j$ are in the same community, $\gamma$ is the resolution parameter, controlling the scale of detected communities.

The Louvain algorithm can be naturally extended to handle bipartite graphs by modifying the modularity function to account for the two disjoint sets of nodes. As proposed by Barber (2007), the bipartite modularity is defined as:

$$Q_b = \frac{1}{m} \sum_{i \in P} \sum_{j \in I} \left[ A_{ij} - \gamma \frac{k_i k_j}{m} \right] \delta(c_i, c_j)$$

where $P$ and $I$ are the two disjoint node sets of the bipartite graph (patients and interactions, respectively, in our case). This formulation ensures that modularity optimization reflects the co-occurrence patterns between patients and shared mechanistic events.

The resolution parameter allows tuning the granularity of the clustering:

- When $\gamma < 1$, the algorithm favors larger communities.
- When $\gamma > 1$, the Louvain method detects smaller clusters.

## Breast cancer use-case

To illustrate the application of *PatientProfiler*, we applied the workflow to a Breast Cancer cohort derived from CPTAC (Krug et al, 2020). The code, the documentation and a tutorial are available at: https://github.com/SaccoPerfettoLab/PatientProfiler_BRCA.

### CPTAC data collection (discovery cohort)

We retrieved Breast cancer data using the *cptac* Python package (version 1.5.0rc1) with the "cptac.download" function and the parameter "dataset = Brca". We downloaded transcriptomic, proteomic, phosphoproteomic, somatic mutation and clinical datasets (Appendix Figs. S2 and S3).

Transcriptomic, proteomic and phosphoproteomic data were analyzed using Step 1 of *PatientProfiler* (*Harmonization of input data*), with default parameters (parameters that differ from default options have been specified below).

Briefly, for transcriptomics we (i) parsed the data; (ii) filtered out rows exceeding the 80% of missing values to remove analyte that are consistently missing in the majority of the patients (Appendix Fig. S3); (iii) performed quality control by removing duplicated entries; (iv) updated gene symbols and added Uni-ProtKB identifiers; (v) computed a sample-wise, median-centered z-score (i.e., based on the distribution of all the genes within a patient/sample).

For proteomics we (i) parsed the data; (ii) performed quality checks by removing duplicated entries; (iii) updated gene names to primary symbols and added UniProtKB identifiers; (iv) imputed missing values using the "norm" method (Bayesian linear regression), instead of the default "pmm" (predictive mean matching), because data is approximately normally distributed; (iv) computed a sample-wise, median-centered z-score.

For phosphoproteomics, we (i) parsed the data; (ii) updated the entry by adding the primary symbols, by adding UniprotKB identifiers, by adding a standardized sequence window, centered on the phosphorylated residue, and by updating the phosphosite position based on the updated UniprotKB reference sequence; (iii) imputed missing values using the "norm" method because data is approximately normally distributed; (iv) computed a sample-wise, median-centered z-score.

Somatic mutation data was annotated as gain-of-function (GOF) or loss-of-function (LOF) using OncoKB API (https://www.oncokb.org/) (RRID:SCR_014782). We used the MafAnnotator.py OncoKB tool via a command line with Python 3 (RRID:SCR_024202) to annotate the impact of each mutation on protein function (1 for GOF and −1 for LOF) (Dataset EV2). We were able to obtain mutation-derived protein activities for 100 out of 122 patients. These were included in the list of patient-specific protein activities measured in the STEP2 of the workflow.

To define patients' subtypes we referred to the Non-Negative Matrix Factorization clustering (NMF.Cluster) described in the publication, from which we extracted the data (Krug et al, 2020).

To generate the collection of mechanistic models and extract the communities, we exploited Step 2, Step 3, and Step 4 of *PatientProfiler*.

Briefly, for Step 2 we used the "extract_cohort_activity" to derive, for each patient, the list of activated and inhibited proteins (final score value). All the chosen parameters, along with the relative explanation have been outlined in Dataset EV1. Briefly, we selected default parameters as discussed in (Venafra et al, 2024), with some exceptions: we didn't use hypergeometric correction in TFEA and KSEA analyses given its high computational cost, also we selected the correct_proteomics = TRUE parameter. This refers to a method that gives less importance to phosphosites equally modulated in proteomics and phosphoproteomics and finds applicability only in KSEA analysis. Next, we integrated patient-specific mutations (Dataset EV2) on the inferred protein activities, overwriting existing proteins or introducing new entries for proteins not originally present in the output.

For Step 3 we used the function "initialize_net_default_params" to set the default parameters and "create_cohort_networks" to iterate

over each patient of the cohort creating the network through the "create_network" function. Specifically, in "create_cohort_networks" we included the following cancer hallmarks as phenotypes: Apoptosis, Autophagy, Cell cycle exit, ECM disassembly, ECM synthesis, Epithelial mesenchymal transition, Immortality, Angiogenesis, Inflammation, Pluripotency, Proliferation, Telomere maintenance. All the selected parameters, along with the relative explanation have been outlined in Dataset EV1. Briefly, we selected default parameters as discussed in (Venafra et al, 2024), with some exceptions, for instance given the prominent presence of transcription factors such as TP53, GATA3 and ESR1 among frequently mutated proteins in breast cancer (Dataset EV2), we decided to set the PKN parameter DIRECT = FALSE, to include also transcriptional regulation.

In Step 4, we used the function "generate_communities" with default parameters. Finally, to extract the transcriptomic signatures, we used patients' communities (or subgroups) and patients' transcriptomic profiles as employed in Step 5 of *PatientProfiler*, with default parameters as detailed in the extended methods section and in Dataset EV1.

### TCGA data collection (validation cohort)

We downloaded transcriptomic and follow-up data for 1094 breast cancer patients from the TCGA using the "TCGAbiolinks package" (version 2.32.0, November 2024) (Koboldt et al, 2012). For the transcriptomic data, we first applied $\log(x + 1)$ transformation and then we applied Step 1 of *PatientProfiler* (*Dataset manipulation)* to filter data and to compute the sample-wise z-score (median-centered). We ranked transcripts in descending order. Next, we used the transcriptomic data to seek for the enrichment of each of the seven transcriptomic signatures (Signatures1–7). To this aim, we performed a gene set enrichment analysis using the "fgseaMultilevel" function with the parameter scoreType = "std" and selected the signatures with lowest adjusted *p*-value with Benjamini-Hochberg (BH) correction. In this way, each patient from TCGA was assigned to a subgroup. Finally, survival curves were generated using the "ggsurvfit" package (version 1.1.0).

To detect whether Signature2- and Signature4-enriched TCGA patients were significantly associated with BL1 and BL2 signatures, we used TCGA patients classification as in (Lehmann et al, 2021), next we perform a Fisher's exact test (BH corrected) to assess the association between our signature-based classification and the previously-characterized Basal-l subtypes.

Reverse phase protein array (RPPA) data from (Koboldt et al, 2012) was downloaded from the TCPA portal at the link, https://tcpaportal.org, selecting Breast cancer data, level 4. Briefly, we retained only data referring to Signature2- and Signature4-enriched patients, next, we only considered analytes present in the graph depicting the functional circuits to key phenotypes in Community 2 and Community 4. Student's t-test (BH corrected) was applied to detect significant modulation between the two groups.

### Covariate analysis

We tested whether CPTAC patients stratified into 7 signatures (Signatures1–7) were associated with the clinical and molecular covariates Stage, Age, Ethnicity, and Tumor Mutational Burden (TMB). Fisher's exact test was used for the analyses, with *p*-values adjusted for multiple testing using the BH correction. Patients were stratified into age categories of <500, 500–700, and >700 months, and into low or high TMB groups based on the median TMB value of the cohort. Only Signature 2 showed a significant positive association with Tumor Mutational Burden (adjusted *p*-value = 0.013). No other covariates (Stage, Age, Ethnicity) were significantly associated with any of the signatures.

### Subtype distribution similarity in CPTAC and TCGA

To compare patient subtype distribution across groups in the CPTAC and TCGA BRCA cohorts, we used the cosine similarity index (Ye, 2011). This metric computes the cosine of the angle between two vectors representing subtype proportions in each group, and ranges from 0 to 1, where 1 indicates identical distributions and 0 divergence. By focusing on the direction rather than the magnitude of vectors, cosine similarity captures how similar subtypes compositions are between cohorts, regardless of different cohort size. This similarity score was used in both randomization and benchmarking analyses.

### Cross-validation with CPTAC dataset

We performed a cross-validation procedure based on random subsampling. Each list of patients of the seven communities was randomly split into a training set (70%) and a test set (30%) across 50 independent iterations. In each iteration, signatures were generated using only the training set and subsequently used to perform enrichment analysis on the test set. We evaluated: (i) signature stability, comparing the gene sets obtained from the training subsets to those derived from the full dataset; and (ii) consistency in patient assignment, by assessing whether test patients were classified into the same communities as in the original run. Precision, Recall, and F1 score were computed for each community, where True Positives (TP) were test patients assigned to the same community as in the original run, False Positives (FP) were assigned to a different community, and False Negatives (FN) were patients not enriched in any signature during cross-validation but assigned in the original analysis.

### Randomization analysis

We generated control signatures by performing gene-wise randomization of the CPTAC transcriptomic data. In this procedure, the expression values of each gene were independently shuffled across samples, preserving the gene-wise distribution while disrupting the sample-wise structure. Randomization was applied to increasing proportions of genes—25%, 50%, 75%, and 100%—across 50 independent iterations for each level. For each randomized dataset, *PatientProfiler* Step 5 was re-run to identify community-specific control signatures. These signatures were then used to perform patient enrichment in the TCGA BRCA cohort. We quantified: (i) the overlap between randomized signatures and the original signatures; (ii) the number of control signatures that enriched patient groups in TCGA; (iii) the prognostic value of enriched groups via Kaplan–Meier survival analysis to five years with adjusted *p*-values (BH correction); and (iv) the cosine similarity of patient subtype distributions between CPTAC and TCGA (see Subtype distribution similarity in CPTAC and TCGA section).

## Over-representation analysis (ORA)

Over-representation analysis was performed using "gprofiler2" software (version 0.2.3) (Kolberg et al, 2020)using the default background and selecting pathway terms from KEGG, Reactome and Wikipathways, biological processes from Gene Ontology and complexes from CORUM (Gene Ontology Consortium et al, 2023; Ashburner et al, 2000; Kanehisa et al, 2023; Gillespie et al, 2022; Martens et al, 2021). Bonferroni-adjusted *p*-values < 0.05 were considered significant. Only top enriched 10 terms were considered.

## Cell cultures

The human breast cancer cell lines MCF7 (RRID:CVCL_0031), MDA-MB-361 (RRID:CVCL_0620) and MDA-MB-231 (RRID:CVCL_0062361) were cultured at 37 °C in a 5% $CO_2$ atmosphere. Mycoplasma contamination was assessed monthly (Abm, US). MCF7 and MDA-MB-231 cells were grown in high-glucose DMEM with sodium pyruvate (Corning, US), containing 10% FBS (Corning, US) and 1 mmol/L l-glutamine (Sigma-Aldrich, DE). MDA-MB-361 cells were grown in RPMI 1640 (Corning, US), containing 10% FBS (Corning, US) and 1 mmol/L l-glutamine (Sigma-Aldrich, DE). Cell lines were not recently authenticated and were tested for mycoplasma contamination.

## Protein purification and quantification

Cell lysis was carried out in RIPA Buffer (ThermoFisher Scientific, US) in the presence of protease inhibitor cocktail (Complete, Roche, US), phosphatase inhibitors (Cocktail Tablets, Roche, US) according to the manufacturer's protocol guidelines. An additional phosphatases inhibitor β-Glycerophosphate (10 mM) (Sigma-Aldrich, DE) was also added to the lysis buffer. Samples have been purified with centrifugation @16,000 × *g* 4 °C for 30 min. Proteins were quantified using the Bradford Protein Assay Reagent (ThermoFisher Scientific, US).

## Western blot

For each sample 20 μg of total protein lysate has been loaded on a denaturing 10% PAGE of acrylamide/bisacrylamide solution 29:1 (Serva, DE). Page Ruler Protein Ladder (ThermoFisher Scientific, US) was used. Proteins were transferred to 0.4 μm nitrocellulose membranes (Bio-Rad, US) at 90 V for 90 min. After the wet transfer, the transfer efficiency was verified using the Ponceau S solution (PanReac, ES, USA). Specifically, membranes were incubated with the ready-to-use Ponceau Red solution for 5 min in agitation at RT, then washed using distilled water to remove the background before the total protein staining (TPS) acquisition. The TPS has been acquired using the "Ponceau S" function of the ChemiDoc MP Imaging System (Bio-Rad). Then, we estimated the total stain volumes per lane using the Image Lab software to get a loading control for quantification. After three washes (10′ 5′ 5′) with 0.1% Tween-20 (Bio-Rad, US) TBS solution (T-TBS), the membranes were blocked using 5% bovine serum albumin (BSA) (Sigma-Aldrich, DE) in T-TBS. The primary Antibodies used are: anti-CDK5 (Cell Signaling, 2506, RRID:AB_2078855) 1:1000 in 5% BSA T-TBS, anti-Phospho-CDK5 (Tyr15) (Thermo, PA577909, RRID:AB_2736361) 1:1000 in 5% BSA T-TBS, anti-Phospho-CDK2

(Thr160) (Cell Signaling, 2561, RRID:AB_2078685) 1:1000 in 5% BSA T-TBS, anti-CDK2 (SantaCruz, sc-6248, RRID:AB_627238) 1:500 in 5% skim milk T-TBS and anti-Cyclin A (B-8) (SantaCruz, sc-271682, RRID:AB_10709300) 1:500 in 5% skim milk T-TBS. Goat anti-rabbit HRP conjugate (Thermo Fisher Scientific) 1:10,000 in 3% skim milk T-TBS and Goat anti-mouse (Thermo Fisher Scientific) 1:10,000 in 3% skim milk T-TBS were used as secondary antibodies. Chemoluminescence was acquired using Clarity western ECL substrate (Bio-Rad, US) in a ChemiDoc MP Imaging System (Bio-Rad, US) (Appendix Figs. S21 and S22). ImageLab Software has been used to quantify the images.

## Data availability

No new experimental data was generated as part of this study. The Breast Cancer datasets are taken from (Krug et al, 2020). Networks generated are available as an NDEx collection at: Patient-specific networks: https://www.ndexbio.org/#/networkset/1387170f-4c20-11f0-a218-005056ae3c32?accesskey=2dfafd3f0a02204ef789f7b4b93741fa193013ae7de552eb60e1ea90e187302a. Communities network: https://www.ndexbio.org/#/networkset/dd734d60-4c24-11f0-a218-005056ae3c32?accesskey=573cb8bc7f0977857ac19a6a6b65c231320fe8baced6a756aaf32570e0a8fa18. PatientProfilerR package code is available at https://github.com/SaccoPerfettoLab/PatientProfiler. The code used for the Breast Cancer use case is available at https://github.com/SaccoPerfettoLab/PatientProfiler_BRCA.

The source data of this paper are collected in the following database record: biostudies:S-SCDT-10_1038-S44320-025-00160-y.

## Peer review information

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

## Acknowledgements

This research was funded by the Italian Association for Cancer Research (AIRC) with a grant to LP (MFAG Grant n. 28858), a grant to FSP (MFAG Grant n. 23099) and a grant to FSA (Start-Up Grant n. 21815). Also, LP and FSA are supported by a joint PRIN 2022 PNRR grant (n. P2022JRETW), funded by the European Union—NextGenerationEU. LP is also supported by a SEED Sapienza Grant. FSA is supported by PRIN 2022 (n. 2022L8RAKN). CC is supported by MUR – PRIN_ 2022 grant (n. 2022ZFFALH; CUP: B53D23003740006). VL is supported by a PNRR fellowship Mission 4, Component 1, Action 4.1 "Research Doctorates", CUP: B53C23001770006 by the European Union-NextGenerationEU. VV is supported by PON-MUR fellowship (n. DOT13IEP1U-1). MLN is supported by MUR - PRIN_2022 (CUP:E53D23009910006). LDR is supported by Sapienza University of Rome (Grant No. RG123188B4885C78) and by INdAM – GNCS Project 2024, CUP_E53C23001670001. EDN was supported by Fondazione Umberto Veronesi.

## Author contributions

**Veronica Lombardi**: Conceptualization; Software; Formal analysis; Investigation; Visualization; Methodology; Writing—original draft. **Lorenzo Di Rocco**: Conceptualization; Software; Methodology; Writing—original draft. **Eleonora Meo**: Formal analysis; Methodology. **Veronica Venafra**: Software; Formal analysis; Writing—original draft. **Elena Di Nisio**: Validation; Visualization; Writing—original draft. **Valerio Perticaroli**: Validation. **Mihail Lorentz Nicolaeasa**: Software. **Chiara Cencioni**: Conceptualization; Writing—review and editing. **Francesco Spallotta**: Conceptualization; Writing—original draft. **Rodolfo Negri**: Validation; Writing—review and editing. **Francesca Sacco**: Conceptualization; Supervision; Writing—review and editing. **Livia Perfetto**: Conceptualization; Supervision; Funding acquisition; Investigation; Methodology; Project administration; Writing—review and editing.

Source data underlying figure panels in this paper may have individual authorship assigned. Where available, figure panel/source data authorship is listed in the following database record: biostudies:S-SCDT-10_1038-S44320-025-00160-y.

## Disclosure and competing interests statement

The authors declare no competing interests.

