## [Peer Review File · Molecular Systems Biology]

PatientProfiler: A network-based approach to personalized medicine

Veronica Lombardi, Lorenzo Di Rocco, Eleonora Meo, Veronica Venafra, Elena Di Nisio, Valerio Perticaroli, Mihail Lorentz Nicolaeasa, Chiara Cencioni, Francesco Spallotta, Rodolfo Negri, Francesca Sacco, and Livia Perfetto

Corresponding author(s): Livia Perfetto (livia.perfetto@uniroma1.it)

Review Timeline:

Submission Date:	23rd Apr 25
Editorial Decision:	28th May 25
Appeal Received:	28th Jul 25
Editorial Decision:	10th Sep 25
Revision Received:	22nd Sep 25
Accepted:	24th Sep 25

Editor: Jingyi Hou

Transaction Report:

26th May 2025

RE: Manuscript MSB-2025-13041, PatientProfiler: A network-based approach to personalized medicine

Dear Dr Perfetto,

Thank you again for submitting your work to Molecular Systems Biology. We have now heard back from the three reviewers who agreed to evaluate your manuscript. As you will see below, the reviewers raise substantial concerns on your work, which unfortunately preclude its publication in Molecular Systems Biology.

While the reviewers acknowledge that the general topic of the study is relevant and important, they raised significant and overlapping concerns about the manuscript. These include issues related to the model validation, benchmarking against relevant methods, and insufficient detail in the description of the proposed approach. Additionally, Reviewer #1 mentioned that the claimed novelty of the work appear overstated, while Reviewer #3 expressed concerns about the analysis conducted in patients and the limited novel insights into cancer offered by the study. Notably, Reviewer #3 rated the general interest and suitability for publication as "low." Given the nature and extent of these concerns, we think the manuscript falls beyond the scope of what could be addressed through a major revision.

However, we would be open to considering a substantially revised and extended version of the manuscript, provided that all the issues raised by the reviewers can be convincingly addressed. We understand if, given the significant revisions required, you may prefer to submit your study elsewhere.

Should you choose to resubmit to us, please note that the manuscript would receive a new number and receipt date. It will be editorially evaluated afresh, and its novelty will be re-assessed at the time of submission. As you probably understand, we can give no guarantee about its eventual acceptability. If you do decide to follow this course, then we would ask you to enclose with your re-submission a point-by-point response to the points raised in the present review.

In the meantime, I discussed your study with Dr. Tim Fessenden, Executive Editor at our not-for-profit, open-access sister journal Life Science Alliance (LSA). Dr. Fessenden expressed interest in your findings and is open to considering the manuscript for publication in LSA, pending the following revisions:

- Provide details on methodology sought by all reviewers.
- Discuss the impact of using patient z-score for making comparisons between patients as noted by Reviewer 2, or clarify that z-scores were not calculated within patients as questioned by Reviewer 3.
- Validate findings using TCGA data to generate control signatures as requested by Reviewer 2.
- Include results of patient stratification based on the 4 classic breast cancer subtypes as requested by Reviewer 3.

We understand that such a revision might need to be re-reviewed, in which case Dr. Fessenden will walk the Reviewers through our transfer process. If you are interested in this option, please use the link below to transfer your manuscript to LSA.

You do not need to revise the manuscript before transferring it to LSA. Once you transfer, Dr. Fessenden will email you an invitation to revise and resubmit, listing the same revision requests as mentioned above. Please feel free to reach out at t.fessenden@life-science-alliance.org if you have any questions about the LSA journal, the transfer process, or the revisions requested.

I am sorry that the review of your work did not result in a more favorable outcome on this occasion, but I hope that you will not be discouraged from sending your work to Molecular Systems Biology in the future. In any case, thank you for the opportunity to examine this work.

Kind regards,
Jingyi

Jingyi Hou, PhD
Senior Editor
Molecular Systems Biology

Reviewer #1:

The manuscript of Lombardi et al introduces PatientProfiler to map how genetic and molecular changes contribute to cancer and other diseases on an individual patient level. Such an approach is timely and important, and the developer group has a great expertise in causal signaling networks.

The manuscript is well written, though often lacks specific details (listed below), the figures are clear and nicely complement the work. The manuscript is complemented by very thorough supplementary information. The documentation and quality of work are excellent.

A main concern with this manuscript is that throughout the text, it overstates the novelty of PatientProfiler, writing it is a "one-of-a-kind tool", while the benchmarking comparison and the literature are not yet supporting this claim. The manuscript misses to compare the PatientProfiler approach to existing relevant tools and works (eg., Pai et al, Molecular Systems Biology, 2019 / Pai et al, F1000R, 2020; Brooks et al, Nature Communications, 2022, Modos et al, medRxiv, 2024; Montagud et al, eLife, 2022). PatientProfiler is different by its integrated set of algorithms, the focus on the effect of coding mutations (as opposed to non-coding regulatory mutations/SNPs) and the disease in which the tool was demonstrated. However, detailed qualitative and, wherever possible, quantitative comparisons are needed to highlight and justify the novelty aspect of PatientProfiler. The current comparison presented in the manuscript compared PatientProfiler with too distant, not that relevant approaches. Likewise, when discussing other tools such as MOFA or CARNIVAL, important downstream steps, like functional annotation of signatures for MOFA or clustering for CARNIVAL, are not mentioned. These steps are typically applied in standard workflows, and omitting them creates an artificial sense of greater difference between these tools and PatientProfiler.

It is just a semantic issue, but the authors did not define what PatientProfiler is, throughout the manuscript it is termed as "tool", "strategy", "algorithm", "workflow", "pipeline". This should be unified and justified. Based on the details provided in the manuscript, PatientProfiler is most likely an integrated pipeline that utilises existing modules, especially the very elegant SignalProfiler, developed by the same group. Therefore, the novelty aspect should be better articulated from this point of view as well.

Additional comments to improve the clarity about the Methods:

- The authors should provide a detailed description of the preprocessing steps applied to both the transcriptomic and proteomic data. They should clarify whether the data were used as-is upon download or subjected to additional processing, and explicitly state all steps taken, including but not limited to Z-score normalization and sequence annotation for phosphoproteomics.
- More explanation is needed for the SignalProfiler package. This should include a clear description of the input data, all parameters used, and how the tool was applied. Additionally, a more detailed depiction of the SignalProfiler workflow should be added to the methodology section, ensuring that the process is understandable even for readers unfamiliar with CARNIVAL or SignalProfiler.
- It would be helpful if the authors could justify their choice of clustering method for the patient networks. They should explain why this approach was selected over alternatives, such as node-specific activity-based clustering. Furthermore, they should provide details on the Louvain clustering method used, including the optimization function, resolution parameters, and how these decisions affect the biological interpretation of the clusters.
- For the Overrepresentation analysis, could the authors explain why they used the very conservative Bonferroni test?

Regarding the Results:

- The authors should include a figure that more clearly illustrates the structure of missing data and the overall composition of the dataset. Specifically, they should clarify the number of features for phosphorylation sites, transcripts, and proteins, and indicate whether the dataset is complete or contains missing values for any samples or feature types.
- Please revise the final sentence of the Breast Cancer Use Case section to adopt a more measured tone. The current wording, implying that the tool correctly captures subtype differences, overstates the results, which simply reflect that normalisation preserves known subtype distinctions. If such differences had not been observed, that would have indicated a serious issue with the normalisation, not necessarily the strength of the tool itself.
- Maybe changing the narrative visual presentation by first presenting the general network, followed by the patient-specific networks could improve clarity. But this is just an optional suggestion.
- Please explain why the communities identified in Figure 4 do not strongly align with clinical phenotypes.
- Replacing the pie charts in Figure 4B with stacked bar plots would better reflect the differences in community size and make it easier for readers to compare proportions across groups.

- Too much relevant information is now presented in the supplementary methods, the authors should consider moving them into the main methods section to support the interpretation of Figure 6 for example. The comparison here needs more explanation as it is not clear completely what was compared to what and how.

Regarding the Discussion section of the manuscript:

- The authors should expand a bit more on the possible reasons for discrepancies between clinical phenotypes and patient-specific network modules. This would strengthen the discussion and provide insight into biological or methodological factors contributing to the findings.
- PatientProfiler is clearly a robust and valuable workflow for patient-specific network analysis. But it is not without precedent. As detailed above, overstating its singularity weakens the credibility of the Discussion.

Reviewer #2:

#Summary

The authors describe their workflow, PatientProfiler, allowing construction of personalised network models for breast cancer, based on a large repository of curated signalling pathways, and a multi-omic dataset. Using PatientProfiler, they analyse data of 122 breast cancer patients and identify seven network signatures characterising these patients. They explore the significance of these signatures for breast cancer subtypes, validate the workflow using an external dataset of over a thousand patients, and compare the performance of the workflow with other methods. Based on their results, they conclude that network-based signatures of cancer signalling support precision medicine as detailed and actionable biomarkers of the disease. The authors share their workflow and its results openly on widely used repositories.

#General remarks

The paper tackles an important and very relevant topic of network-level signatures in complex diseases, where relationships between individual biomarkers may bring new insights into disease mechanisms, and improve the precision of diagnosis. In this context, PatientProfiler advances the field of cancer diagnostics by proposing a new computational workflow with potential clinical relevance. The use of network biomarkers will be appealing to the bioinformaticians and computational biologists, however clinicians may also find this article appealing because of visual representation of relationships between individual biomarkers.

Given the above, the article requires additional details regarding the workflow and the validation of the results. First, insufficient description of the workflow components, their impact on the output, and their parameterisation makes it difficult to properly evaluate the novelty of the workflow. Second, the authors decide to calculate individual omics signatures based on z-scores, which does not provide a frame of reference between patients, or comparison to control samples. The impact of this decision is not evaluated. Third, validation of the results should be more detailed, including testing how robust the signatures are, and their specificity to the source dataset, beyond Kaplan-Meier curves and enrichment analysis.

#Major points

##Clarity of algorithm description and parameterisation

PatientProfiler workflow is composed of multiple components, some of them are already published and used independently (CARNIVAL, SignalingProfiler). What is the impact of this previous work on the overall performance of PatientProfiler? Functionalities of SignalingProfiler, like PhenoScore algorithm and "footprint-based methods", are mentioned but not explained.

In Step 2-3 (page 3), "five ad-hoc developed functions" are mentioned. Their role in the workflow should be described.

In Step 5 (page 4), "signatures from the obtained communities" are mentioned. Communities are clustered in Step 4. Does this part refer to the communities or clusters? Similarly, in paragraph 4, page 5 "patients' communities (or subgroups)" are mentioned, which is unclear.

A number of workflow parameters are suggested without explaining their impact on the outcome. These include:

- "maximum percentage of missing data are removed" (page 3)
- "+/-2 standard deviations" as a significance criterion (page 3)
- "with default SignalingProfiler 2.0 parameters" (page 3)
- initialize_net_default_params function, which automatically sets default parameters (page 3)

- "a minimum expression difference of 0.7" (page 4),
- "Only for Signature 1, 5 and 6 we selected a minimum expression difference of 0.5" (page 4)
- paragraph 4, page 5

##Calculation of molecular activity

Patient-specific profiles are calculated without a frame of reference to other patients or healthy controls. The authors claim that this analysis is unbiased, however it makes it relative to the in-sample variance. Replacement of log-fold change with a z-score indeed allows to investigate patients individually, but the authors should discuss the impact of this decision on downstream analysis, and how it affects the robustness of the signatures derived in such a way. Supplementary Figure 12 suggests that this can be a problem: one of the signatures is not identified in the TCGA dataset, and there is a high variability between patients classified by other signatures, bigger than in the source dataset.

Also, the authors should explain the choice of phenotypes in the networks. In my understanding, SIGNOR features more phenotypes than only those reported in NDex network. In the article, the authors refer to the hallmarks of cancer, but the mapping is not explicitly provided. In some of the networks shared via NDex there's "Brown adipogenesis" phenotype, is this intended?

##Improve validation of the network models

Network models introduced in the article are an output of a complex workflow, and validation of their robustness is recommended, especially if they are foreseen as a support in diagnostics or therapy. Approaches like re-running the workflow with cross-validation for the same dataset, or generating control signatures after randomly perturbing expression signatures in samples will provide insight of how specific networks are to the data used to construct them.

Validation using the TCGA dataset was done via enrichment against already computed network signatures. An expected validation would be to use TCGA data to generate control signatures and compare them with those from CPTAC dataset. This is key, as validation via enrichment using TCGA data, as well as comparison with other tools, assume that the signatures produced by PatientProfiler are correct.

Comparison with other tools relies on the results of PatientProfiler, which may bias the results, as other methods are asked to recreate output of the workflow. Rationale about seven clusters being the value of choice (Extended methods, page 2) may be incorrect, as clustering is applied on different stages of data processing - in PatientProfiler it happens on the network level, while other methods use source data. Moreover, interpretation of Figure 6B seems to be imprecise. The authors say "Furthermore, when we focus on signatures enriching Basal-I patients exclusively, only PatientProfiler and tSNE+KMeans produce groups with clinically relevant survival differences." While Fig 6B shows PCA+hierarchical clustering outperforming PatientProfiler for "all signatures" category, and MOFA+ hierarchical clustering outperforming it for "basal-specific signatures". Also, values in Figure 6A do not correspond to Supplementary Figure 15 (MOFA+ hierarchical clustering for basal-like has $p=0.9$ there, which should give $-\text{Log}(p) = 0.11$).

#Minor points

PatientProfiler has a minor installation error: SignallingProfiler is not listed in dependencies, separate installation of SignallingProfiler allowed to install the workflow

Clarity of explanations across the article can be improved, including:

- Introduction, page 2, paragraph 3, sentences starting from "Consequently" are incoherent.
- Methods, workflow description, e.g. last paragraph of Step 1

The authors should consistently use of one word describing PatientProfiler (currently strategy, pipeline, workflow, tool), workflow is suggested

"Importantly, applying the analyte-wise (by row) and/or the sample-wise (by column) z-score." (page 3) this sentence sounds broken

Typos found in the article

- "unbiasedly" should be "in an unbiased way" (several places)
- "from STEP 3" should be from "STEP 2" (Fig 1 caption)
- "rank" should be "ranked" (page 5)
- "jupyter" should be "python" (Fig 2 caption,)
- "phosphates" (page 6)

Reviewer #3:

This study seeks to identify patient-specific models and breast cancer subtypes by combining multi-omic data with established signaling networks. The approach first infers protein activities from transcriptomic and phosphoproteomic data, then uses these

predicted activities to define patient-specific signaling networks. These patient networks are clustered to identify seven breast cancer subtypes which are examined to understand the activation of cancer hallmark pathways and differences in patient prognosis. The study suggests that the Basal-like breast cancer subtype may be made up of two distinct signatures, with differences in prognosis.

The rationale for examining differential signaling and regulatory networks across cancer patients to define clinical subtypes and promote precision medicine is sound. However, a number of methodological aspects are unclear, several conclusions are inconsistent with the data presented, and some conclusions are overstated. The findings are largely consistent with known cancer biology and patient outcomes, such as poorer survival outcomes for patients with TNBC, but more work is needed to convincingly add to the current state of cancer knowledge. For example, Wang 2019 previously identified two basal-like subtypes, and the prognostic differences observed for other signatures in this study appear minimal. With significant refinements, this study could interest those working on tumor signaling and network approaches for multi-omic analysis.

Specifically:

The analysis does not account for important covariates, including age or tumor mutational burden.

There is an over-reliance on the predictions of the 4 classical subtypes (Basal-I, Her2-I, LumA-I, and LumB-I) across the later analyses. This is confusing given the study premise that these 4 subtypes are insufficient for classifying patients, and are themselves a computational classification with potential for noise. Further, the patient stratification results are not provided for the 4 classical subtypes, leaving the benefit of additional subtypes unclear.

The relationship of the results to the genomic profiles of each patient is unclear (is genomic information incorporated in PatientProfiler?), and limits the utility in a clinical setting. This genomic information could be utilized to measure and understand the coherence of the proposed subtypes.

The methodology underlying PatientProfiler lacks some details and could be streamlined to reduce repetition in the methods, supplemental methods, results, and discussion:

(Step 1) What, if any, imputation was performed? Please clarify whether "Sample-wise z-scoring" is Z-scoring of each gene across all patients or all genes within a patient.

(Step 2) Please provide further details on the protein inference methods (e.g. VIPER) and rationale for parameter choices (e.g. `correct_proteomics = FALSE` for TFEA when the SignalingProfiler publication states that setting this to `TRUE` reduces RMSE)

(Step 4) The schematic implies that the bipartite community detection partitions all patients and proteins into a single community each. However, numerous proteins are present in multiple communities (E.g. MAPK1 in 2 & 4) - how similar are the different communities? How should one interpret the presence of both an up-regulated and down-regulated version of a protein in a community network (e.g. MAPK1 in community 6)? Could the confidence of the patient-community assignments be quantified?

(Step 4) What are the impacts (e.g. proportion of patient networks excluded) for each of the filtering steps applied to G*? In Breast Cancer use case, it is stated that "mutation in TP53 seems to associate prevalently with LumA-I and Basal-I patients." However, SFig 1 shows only 4/25 LumA-I patients with TP53 mutations. Should this state 'HER2-I', which would be consistent with Krug, 2020? Further, it does not appear that all 122 patients are present in the analysis in SFig 1 - why is this?

In Protein activity estimation, the statement "we observe that CDK2 and CDK5 are more active in Basal-I cell line as compared to cell lines representative of other subtypes" is inconsistent with the data presented. Additionally, the hypothesis being tested in this experiment (and its novelty given prior work on CDKs in TNBC, such as 1 and 2) is unclear, and key methodological details and justifications are missing: How many replicates were performed? How was TPS quantified? What is the justification for using TPS rather than a more typical loading control, such as Actin? Has the TPS image been cropped (One would expect detection of protein bands outside the range shown)?

In Application: Communities' prioritization and transcriptome signatures extraction. Differences in phenoscores should be quantified statistically to support proposed biological differences between the subgroups. Overrepresentation analysis uses the entire human proteome as background (SFig 9), but PatientProfiler only has access to 7,500 proteins (Discussion). Therefore, the enrichments could simply reflect known biases of the underlying network. Why are only up-regulated transcripts considered when defining the transcriptomic signatures?

In Signature validation on an independent Breast Cancer cohort: please provide sample size information for the survival curves. In Figure 5B, there appear to be only a few patients represented by signature 4. Are these patients actually representative of the signature?

In Comparison with other methods: The recasting of p-values as a 'similarity index' with an emphasis on comparison of non-significant values is confusing, especially as a non-significant p-value could result from an underpowered test rather than true

similarity. One idea for a more robust approach would be to measure the similarity of genomic or transcriptomic profiles of patients assigned to each cluster. Also, it is stated that "PatientProfiler and tSNE+KMeans produce groups with clinically relevant survival differences", yet Figure 6B does not show any significant survival differences for tSNE+KMeans. How were the basal-specific signatures chosen? Were there exactly two basal-enriched subtypes from each method, or were the top two most enriched subtypes chosen? Was multiple-hypothesis correction performed?

Minor points

"Workflow designed to unbiasedly integrate cancer-derived data" - what is meant by 'unbiasedly' in this context? One could argue that the choice of a signaling network and the filtering criteria imposed on the structure of G* introduce biases into the analysis.

Please specify all statistical tests performed by name, and specify the multi-hypothesis correction method.

Please check the significance designations in SFig 3. E.g., for TOP2A it seems unlikely that the difference between LumA-I and LumB-I is more significant than the difference between Her2-I and LumB-I.

Please check all figure references and panel labels for accuracy. E.g., A 'Figure 3D' is referenced but does not exist.

Network diagrams lack a visual legend

In Figure 3A, alignment of the y-axes would enable better comparison between groups.

Typos in SFig 6: Panel C is not indicated, and the ladder for pTyr15-CDK5 is marked at 35kDa.

Methods Patient Profiler Step 4 uses the term 'interaction' to refer to nodes in the bipartite graph, which leads to confusion as 'interaction' is typically used to refer to the edges. 'Interactor' or simply 'node' would be preferred.

Please explain the inconsistency in the number of nodes present in Figure 3A and the number of nodes per patient in Figure 3B (which specifies a minimum of ~100).

The discussion should clearly summarize and contextualize the mechanisms underlying differences across patients.

The availability of the pipeline code and all generated networks is appreciated.

** As a service to authors, EMBO Press offers the possibility to directly transfer declined manuscripts to another EMBO Press title or to the open access journal Life Science Alliance launched in partnership between EMBO Press, Rockefeller University Press and Cold Spring Harbor Laboratory Press. The full manuscript and if applicable, reviewers' reports, are automatically sent to the receiving journal to allow for fast handling and a prompt decision on your manuscript. For more details of this service, and to transfer your manuscript please click on Link Not Available. **

Reviewer #1

The manuscript of Lombardi et al introduces PatientProfiler to map how genetic and molecular changes contribute to cancer and other diseases on an individual patient level. Such an approach is timely and important, and the developer group has a great expertise in causal signaling networks.

The manuscript is well written, though often lacks specific details (listed below), the figures are clear and nicely complement the work. The manuscript is complemented by very thorough supplementary information. The documentation and quality of work are excellent.

POINT 1. A main concern with this manuscript is that throughout the text, it overstates the novelty of PatientProfiler, writing it is a "one-of-a-kind tool", while the benchmarking comparison and the literature are not yet supporting this claim. The manuscript misses to compare the PatientProfiler approach to existing relevant tools and works (eg., Pai et al, Molecular Systems Biology, 2019 / Pai et al, F1000R, 2020; Brooks et al, Nature Communications, 2022, Modos et al, medXriv, 2024; Montagud et al, eLife, 2022). PatientProfiler is different by its integrated set of algorithms, the focus on the effect of coding mutations (as opposed to non-coding regulatory mutations/SNPs) and the disease in which the tool was demonstrated. However, detailed qualitative and, wherever possible, quantitative comparisons are needed to highlight and justify the novelty aspect of PatientProfiler. The current comparison presented in the manuscript compared PatientProfiler with too distant, not that relevant approaches. Likewise, when discussing other tools such as MOFA or CARNIVAL, important downstream steps, like functional annotation of signatures for MOFA or clustering for CARNIVAL, are not mentioned. These steps are typically applied in standard workflows, and omitting them creates an artificial sense of greater difference between these tools and PatientProfiler.

We thank the reviewer for carefully reading the manuscript, for his/her/their encouraging words and for the proposed improvements. As the reviewer mentioned, *PatientProfiler* represents a unique set of algorithms that cover many aspects: patients-specific network generation, data reduction, clustering, and biomarkers identification. We appreciated that the reviewer understood that our "one-of-a-kind tool" was referring to that. However, we understand that, for any of the aspects covered (patients-specific network generation, data reduction, clustering, and biomarkers identification), there are other tools that can tackle similar challenges. To avoid overstatements, we have removed the "one-of-a-kind tool" sentence and carried out an extensive revision of the discussion paragraph. Also, we significantly expanded the "comparison with other tools" paragraph to ensure a fairer description of other methods (such as pCHIPs, COSMOS and MOFA). Finally, we also included additional tools such as iSNP, PROFILE and NetDx in our qualitative comparisons. This expanded comparison highlights that the field is rapidly evolving, and that each method addresses distinct research questions. Nevertheless, it is important to highlight that some tools reviewed here were

developed for different, complementary, purposes. As an example, PROFILE has been used to predict patient-specific response to combined therapeutic strategies, MOFA has been perfected to identify factors that are associated with existing groups and for stratification, COSMOS+ to bridge mechanistic hypothesis generation with factor analysis, iSNP to map the functional impact of SNPs and identify areas of a global regulatory network impacted by these vulnerabilities or netDx predicts risk assessment. Given these differences in scope and functionality, we believe that a quantitative comparison may not be appropriate and could potentially lead to misleading conclusions and overlook the distinct features of each tool.

As such, we have decided to maintain only a qualitative comparison between tools. We have added a short paragraph in the main text (Pag. 19) and summarized the comparison in **Supplementary Figure 20**.

Additional points raised have been discussed in this rebuttal. Below are our point-by-point replies.

It is just a semantic issue, but the authors did not define what PatientProfiler is, throughout the manuscript it is termed as "tool", "strategy", "algorithm", "workflow", "pipeline". This should be unified and justified. Based on the details provided in the manuscript, PatientProfiler is most likely an integrated pipeline that utilises existing modules, especially the very elegant SignalProfiler, developed by the same group. Therefore, the novelty aspect should be better articulated from this point of view as well.

As pointed out by reviewers 1 and 2, there was a misuse of terms. We established it to refer to *PatientProfiler* as a "workflow". Text has been changed accordingly.

Additional comments to improve the clarity about the Methods:

##Z-score

POINT 2. The authors should provide a detailed description of the preprocessing steps applied to both the transcriptomic and proteomic data. They should clarify whether the data were used as-is upon download or subjected to additional processing, and explicitly state all steps taken, including but not limited to Z-score normalization and sequence annotation for phosphoproteomics.

To address reviewer concerns, we have now detailed in the methods the processing steps accordingly (Pag 3).

POINT 3. More explanation is needed for the SignalProfiler package. This should include a clear description of the input data, all parameters used, and how the tool was applied. Additionally, a more detailed depiction of the SignalProfiler workflow should be added to the

methodology section, ensuring that the process is understandable even for readers unfamiliar with CARNIVAL or SignalProfiler.

We totally acknowledge the reviewer's point; we have now extensively expanded the methods to better explain the *SignalingProfiler 2.0* package. Here each parameter has been described, with clarification about its usage within *PatientProfiler* (new **Supplementary Table 1**). In addition, we also added a novel **Supplementary Figure 1** that illustrates in detail the steps of *SignalingProfiler 2.0*.

POINT 4. It would be helpful if the authors could justify their choice of clustering method for the patient networks. They should explain why this approach was selected over alternatives, such as node-specific activity-based clustering. Furthermore, they should provide details on the Louvain clustering method used, including the optimization function, resolution parameters, and how these decisions affect the biological interpretation of the clusters.

We understand and we apologize for the lack of information.

Our choice of not adopting node-specific activity-based clustering depends on the intent of our analysis. Our objectives included the possibility to derive mechanistic insights into the dysregulations that characterize each stratification group, which cannot be derived exclusively from gene activity, but from the way these genes interact within the broader network context.

For this reason, we opted to generate a bipartite graph representing patient-to-interaction relationships and applied the Louvain algorithm. Specifically, the Louvain algorithm offers a scalable solution for partitioning large graphs into communities by maximizing modularity, without requiring the number of clusters to be specified in advance. In our context, this strategy allows for the grouping of patients based on shared mechanistic patterns inferred from their interactomes, rather than relying solely on individual node activity.

When using the Louvain algorithm, the resolution parameter is pivotal, as it determines the granularity of the detected communities. Larger resolution values result in more, smaller clusters, whereas smaller values yield fewer, broader communities. By varying this parameter, we can control the level of granularity, which directly impacts how patient subtypes are defined. We tuned the resolution parameter empirically by repeating the clustering process and evaluating a range of values. Then, we selected the value leading to the partition with the highest modularity score.

Overall, to enhance the clarity of our work and address reviewer's point:

1. We have now added a dedicated paragraph in the Extended Methods providing a more detailed explanation of how the Louvain algorithm works, including its parametrization.
2. We have now included the new **Supplementary Figure 11A** representing our empirical analysis for the choice of the resolution parameter.

New Supplementary Figure 11A Tuning of Resolution Parameter. Effect of resolution parameter (γ) on community detection. To empirically tune the resolution parameter, we performed community detection across a range of resolution values and evaluated the resulting partitions based on their modularity scores. The number of detected communities (left y-axis, blue line) increases with γ , while the modularity score (right y-axis, red dashed line) peaks around 1.0. The resolution value corresponding to the highest modularity score was selected for downstream analysis ($\gamma=0.99$).

3. In the results and in the discussions sections, we have provided the rationale behind our decisions and discussed the possibility to implement alternative clustering methods.

POINT 5. For the Overrepresentation analysis, could the authors explain why they used the very conservative Bonferroni test?

We decided to use Bonferroni as there is no assumption of dependency and works well under both independent and dependent testing scenarios.

Regarding the Results:

POINT 6. The authors should include a figure that more clearly illustrates the structure of missing data and the overall composition of the dataset. Specifically, they should clarify the number of features for phosphorylation sites, transcripts, and proteins, and indicate whether the dataset is complete or contains missing values for any samples or feature types.

We agree with the reviewer, and we have now added a preliminary presentation of the datasets. The total number of analytes in each omic layer, together with the number of missing values in each sample/patient and the percentage of missing values for each analyte in the 122 patients, are now provided in the figure below (new **Supplementary Figure 3**).

New Supplementary Figure 3. Missing values in CPTAC datasets. A) Distribution of missing values within the same sample (patient), in parenthesis the total number of analytes in each omic level. **B)** Percentage of missing values within the same analyte, the total number of patients is 122.

POINT 7. Please revise the final sentence of the Breast Cancer Use Case section to adopt a more measured tone. The current wording, implying that the tool correctly captures subtype differences, overstates the results, which simply reflect that normalisation preserves known subtype distinctions. If such differences had not been observed, that would have indicated a serious issue with the normalisation, not necessarily the strength of the tool itself.

We have now rephrased the sentence to adopt a more measured tone (Pag 18).

POINT 8. Maybe changing the narrative visual presentation by first presenting the general network, followed by the patient-specific networks could improve clarity. But this is just an optional suggestion.

We thank the reviewer for this suggestion. However, in our current approach, we do not generate a general (aggregate) network, as our method directly generates individualized patient-specific mechanistic models. In this perspective, reordering the visual presentation in that way is not applicable in our case.

POINT 9. Please explain why the communities identified in Figure 4 do not strongly align with clinical phenotypes.

We thank the reviewer for this comment, we have now added a novel paragraph to provide a deeper discussion of the detected communities. Also, we have also introduced text to explain why our network-based communities do not strongly align with clinical phenotypes (we assume here the reviewer refers to classical subtypes, namely LumA-I, LumB-I, HER2-I, Basal-I). *Our community-based stratification, derived from mechanistic multi-omic modeling,*

partially overlaps with intrinsic subtypes (as shown by enrichment analysis) and captures orthogonal features. This discrepancy might be due to the limited capabilities of PatientProfiler (discussed below), but it could also mirror the inadequacy of the current classification in capturing the complexity and the heterogeneity of Breast Cancer patients and might reflect the added resolution provided by our integrative approach.

POINT 10. Replacing the pie charts in Figure 4B with stacked bar plots would better reflect the differences in community size and make it easier for readers to compare proportions across groups.

Bar plots were generated and replaced the pie charts in **Figure 4B** and in relevant Supplementary Figures.

POINT 11. Too much relevant information is now presented in the supplementary methods, the authors should consider moving them into the main methods section to support the interpretation of Figure 6 for example. The comparison here needs more explanation as it is not clear completely what was compared to what and how.

As discussed in POINT 1, the quantitative comparison between tools has been withdrawn from the analysis.

Regarding the Discussion section of the manuscript:

POINT 12. The authors should expand a bit more on the possible reasons for discrepancies between clinical phenotypes and patient-specific network modules. This would strengthen the discussion and provide insight into biological or methodological factors contributing to the findings.

As previously discussed, we have now expanded the discussion accordingly.

POINT 13. PatientProfiler is clearly a robust and valuable workflow for patient-specific network analysis. But it is not without precedent. As detailed above, overstating its singularity weakens the credibility of the Discussion.

We understand and agree. As discussed in a previous point (See POINT 9), we have revised the discussion, presented similar tools, and adopted a more measured tone.

Reviewer #2

The authors describe their workflow, PatientProfiler, allowing construction of personalised network models for breast cancer, based on a large repository of curated signalling pathways,

and a multi-omic dataset. Using PatientProfiler, they analyse data of 122 breast cancer patients and identify seven network signatures characterising these patients. They explore the significance of these signatures for breast cancer subtypes, validate the workflow using an external dataset of over a thousand patients, and compare the performance of the workflow with other methods. Based on their results, they conclude that network-based signatures of cancer signalling support precision medicine as detailed and actionable biomarkers of the disease. The authors share their workflow and its results openly on widely used repositories.

The paper tackles an important and very relevant topic of network-level signatures in complex diseases, where relationships between individual biomarkers may bring new insights into disease mechanisms, and improve the precision of diagnosis. In this context, PatientProfiler advances the field of cancer diagnostics by proposing a new computational workflow with potential clinical relevance. The use of network biomarkers will be appealing to the bioinformaticians and computational biologists, however clinicians may also find this article appealing because of visual representation of relationships between individual biomarkers.

We thank the reviewer for his/her/their careful revision and for the encouraging words.

POINT 14. Given the above, the article requires additional details regarding the workflow and the validation of the results. First, insufficient description of the workflow components, their impact on the output, and their parameterisation makes it difficult to properly evaluate the novelty of the workflow. Second, the authors decide to calculate individual omics signatures based on z-scores, which does not provide a frame of reference between patients, or comparison to control samples. The impact of this decision is not evaluated. Third, validation of the results should be more detailed, including testing how robust the signatures are, and their specificity to the source dataset, beyond Kaplan-Meier curves and enrichment analysis.

We understand and agree with the reviewer's points. We have now revised the manuscript at the text, figure and method levels and the supplementary methods to provide additional parameters and details about the workflow and the rationale for parametrization; we discussed the z-scoring choice; finally, we implemented robustness and cross-validation analyses to assess our methods, beyond Kaplan-Meier curves and enrichment analysis (see **Annex 1**). We are sincerely thankful for the comments, and we strongly believe that these additional analyses have significantly improved the quality of our work.

All the points raised have been discussed in this rebuttal. Below are our point-by-point replies.

#Major points

##Clarity of algorithm description and parameterisation

We apologize for the lack of information; we have now extensively expanded this part in the extended methods.

POINT 15. PatientProfiler workflow is composed of multiple components, some of them are already published and used independently (CARNIVAL, SignalingProfiler). What is the impact of this previous work on the overall performance of PatientProfiler? Functionalities of SignalingProfiler, like PhenoScore algorithm and "footprint-based methods", are mentioned but not explained.

We totally acknowledge the reviewer's point; we have now extensively expanded the extended methods to better explain components from other packages such as CARNIVAL and SignalingProfiler 2.0. Here each parameter has been described, with clarification about its usage within PatientProfiler (also summarised in new **Supplementary Table 1**). In addition, we also added a novel **Supplementary Figure 1** that illustrates in detail the steps of SignalingProfiler 2.0, including PhenoScore, PhosphoScore and footprint-based methods.

POINT 16. In Step 2-3 (page 3), "five ad-hoc developed functions" are mentioned. Their role in the workflow should be described.

We apologize for the lack of information; we have now extensively expanded this part in the methods section and in **Supplementary Table 1** (now in the extended methods).

POINT 17. In Step 5 (page 4), "signatures from the obtained communities" are mentioned. Communities are clustered in Step 4. Does this part refer to the communities or clusters? Similarly, in paragraph 4, page 5 "patients' communities (or subgroups)" are mentioned, which is unclear.

"Community" and "cluster" terms are often used as synonyms in graph theory applications.

POINT 18. A number of workflow parameters are suggested without explaining their impact on the outcome. These include:

- "maximum percentage of missing data are removed" (page 3)

This has now been specified in text.

- "+/-2 standard deviations" as a significance criterion (page 3)

"±2 standard deviation" (z-score ± 1.96) from the mean/median is considered the standard threshold for significance in z-score. To clarify this, we have revised the text as follows: "Values greater or less than ±1.96 (i.e., ±2 standard deviations) are considered significantly modulated, as they fall outside the 95% confidence interval (p-value < 0.05) (Cheadle et al, 2003)"

- "with default SignalingProfiler 2.0 parameters" (page 3)

Default parameters of *SignalingProfiler* 2.0 have been systematically defined and described in the original publication (Venafrà et al, 2024) and recapitulated in the extended methods and in new **Supplementary Table 1**.

- initialize_net_default_params function, which automatically sets default parameters (page 3)

We apologize for the lack of information; we have now extensively expanded this part in the extended methods.

- "a minimum expression difference of 0.7" (page 4), "Only for Signature 1, 5 and 6 we selected a minimum expression difference of 0.5" (page 4)

We apologize for the lack of clarity; the minimum expression difference depends on the average difference in the signature expression. This has now been clarified in methods and in the code documentation.

- paragraph 4, page 5

Z-score, Calculation of molecular activity

POINT 19. Patient-specific profiles are calculated without a frame of reference to other patients or healthy controls. The authors claim that this analysis is unbiased, however it makes it relative to the in-sample variance. **Replacement of log-fold change with a z-score indeed allows to investigate patients individually, but the authors should discuss the impact of this decision on downstream analysis, and how it affects the robustness of the signatures derived in such a way.** Supplementary Figure 12 suggests that this can be a problem: one of the signatures is not identified in the TCGA dataset, and there is a high variability between patients classified by other signatures, bigger than in the source dataset.

We totally understand the reviewer and the editor' concerns. The decision to use sample-wise z-scoring in *PatientProfiler* was driven primarily by the nature of the available data, since sample data is usually present as a single replicate (n =1) and matched normal tissues are not available for all the cohorts, as is the case of our breast cancer use case, impeding standard log-fold change computations. We acknowledge that this makes it difficult to define expression changes relative to a biological baseline. However, we believe that sample-wise z-scoring remains a reasonable approach. Importantly, intra-sample z-scoring has been previously proposed (Krug et al, 2020). Additionally, we have performed a new analysis to assess the consistency of our approach and its applicability in cross-cohort comparison. Specifically, we analysed the z-score profiles of 20 well-established breast cancer biomarkers reported in (Nolan et al, 2023). As shown in the figure below (**new Supplementary Figure 4**), the trends we observe for these genes in both CPTAC and TCGA datasets are very similar, supporting that our intra-sample normalization preserves meaningful biological signals across cohorts.

Figure_A Picture from Nolan et al, 2023. **Expression heatmap of breast cancer biomarkers in TCGA’s zscored transcriptomics data.** Gene expression data for key genes associated with the breast cancer intrinsic subtypes. Breast tumor samples from the TCGA (n = 792) were ordered according to the main intrinsic subtypes for selected genes. Yellow, higher than median gene expression; black, median; blue, lower than median. **Figure_B (New Supplementary Figure 4) Expression heatmap of breast cancer biomarkers in CPTAC’s zscored transcriptomics data.** The heatmap shows distinct expression patterns across different subtypes (LumA-I, LumB-I, HER2-I, Basal-I) consistent with those observed in the TCGA dataset Nolan et al, 2023 and Hoadley et al. 2014 (in **A**).

To conclude, we believe that sample-wise z-scoring is a practical strategy for the analysis of patient-specific omics profiles in the absence of normal tissue score controls and/or in absence of more replicates. It allows for signature detection across datasets and for comparison between independent cohort.

As mentioned by the reviewer, one of the signatures is not identified in the TCGA dataset. Our data support the hypothesis that this could be mainly due to heterogeneity at the level of community 6 which also appears strongly affected by randomization (See **Annex 1**). Prompted by the reviewer comment, we have been extensively results and discussion and integrated novel a new **Supplementary Figure 4**.

POINT 20. Also, the authors should explain the choice of phenotypes in the networks. In my understanding, SIGNOR features more phenotypes than only those reported in NDex network. In the article, the authors refer to the hallmarks of cancer, but the mapping is not explicitly provided. In some of the networks shared via NDex there's "Brown adipogenesis" phenotype, is this intended?

The complete list of cancer hallmark phenotypes includes Apoptosis, Autophagy, Cell cycle exit, ECM disassembly, ECM synthesis, Epithelial mesenchymal transition, Immortality, Angiogenesis, Inflammation, Pluripotency, Proliferation, Telomere maintenance. We apologize for the inconvenience, “Brown adipogenesis” was not meant to be incorporated as a cancer hallmark. We have now corrected the mistake and generated the correct networks and made these available in NDEx. The complete list of chosen phenotypes has now been

added to methods. To note, *PatientProfiler* integrated only a limited number of phenotypes in the maps, as indicated in **Figure 3C** of the manuscript.

##Improve validation of the network models

POINT 21. Network models introduced in the article are an output of a complex workflow, and validation of their robustness is recommended, especially if they are foreseen as a support in diagnostics or therapy. Approaches like re-running the workflow with cross-validation for the same dataset, or generating control signatures after randomly perturbing expression signatures in samples will provide insight of how specific networks are to the data used to construct them.

We understand these were major points of weakness of our work. To address the reviewer's point, we have generated new analyses, which have now been presented in a novel paragraph and in **Figure 6** and **New Supplementary Figures 19**. The **Annex 1 "Validation of the network models"** section, at the end of this rebuttal, recapitulates the analyses and relative results. Collectively, these results reinforce the reliability and robustness of *PatientProfiler*. We are sincerely grateful to the reviewer for highlighting this issue, which we believe has led to a substantial improvement in the quality of our work.

POINT 22. Validation using the TCGA dataset was done via enrichment against already computed network signatures. An expected validation would be to use TCGA data to generate control signatures and compare them with those from CPTAC dataset. This is key, as validation via enrichment using TCGA data, as well as comparison with other tools, assume that the signatures produced by *PatientProfiler* are correct.

We agree with the reviewer that using TCGA data to generate control signatures and compare them with those from CPTAC dataset would represent the ideal validation. However, TCGA data do not include bulk proteomic and phosphoproteomic data. Therefore, using TCGA would entail working with different data types, potentially introducing biases due to differences in data input rather than reflecting true biological variation. Consequently, we cannot carry out the suggested analysis. However, we sincerely hope that the reviewer will agree with us that the randomization and cross-validation results (**See Annex 1, "Validation of the network models"**) can provide sufficient validation of the network/signatures. Additionally, we want to highlight that our strategy identified two signatures (Signature2- and Signature4) that were significantly associated with the well-characterized BL1 and BL2 signatures (Lehmann *et al*, 2011, 2016; Wang *et al*, 2019).

Comparison with other tools

POINT 23. Comparison with other tools relies on the results of *PatientProfiler*, which may bias the results, as other methods are asked to recreate output of the workflow. Rationale about seven clusters being the value of choice (Extended methods, page 2) may be incorrect, as clustering is applied on different stages of data processing - in *PatientProfiler* it happens on the network level, while other methods use source data.

We thank the reviewer for pointing out this potential source of bias. As extensively mentioned in the reply to reviewer 1 (please refer to POINT 1), we have decided to remove the quantitative comparison with other tools, and to limit the comparison to qualitative differences.

POINT 24. Moreover, interpretation of Figure 6B seems to be imprecise. The authors say "Furthermore, when we focus on signatures enriching Basal-I patients exclusively, only PatientProfiler and tSNE+KMeans produce groups with clinically relevant survival differences." While Fig 6B shows PCA+hierarchical clustering outperforming PatientProfiler for "all signatures" category, and MOFA+ hierarchical clustering outperforming it for "basal-specific signatures". Also, values in Figure 6A do not correspond to Supplementary Figure 15 (MOFA+ hierarchical clustering for basal-like has $p=0.9$ there, which should give $-\text{Log}(p) = 0.11$).

We apologize for the confusion, in the submitted version of the manuscript we forgot to update old Figure 6. However, as mentioned in the previous point (please refer to POINT 1), we have decided to censor the quantitative comparison with other tools, and to limit the comparison to qualitative differences.

#Minor points

POINT 25. PatientProfiler has a minor installation error: SignallingProfiler is not listed in dependencies, separate installation of SignallingProfiler allowed to install the workflow

SignalingProfiler is not currently listed among the dependencies because it has not yet been included in Bioconductor. We are actively working on this development. Once this will be concluded, it will be our top priority to add *SignalingProfiler* to the dependencies. We added a recommendation in the GitHub to first install *SignalingProfiler*.

POINT 26. Clarity of explanations across the article can be improved, including:

- Introduction, page 2, paragraph 3, sentences starting from "Consequently" are incoherent.

"Consequently" has been removed.

- Methods, workflow description, e.g. last paragraph of Step 1. The authors should consistently use of one word describing PatientProfiler (currently strategy, pipeline, workflow, tool), workflow is suggested

As pointed out by reviewers 1 and 2, there was a misuse of terms. We established it to refer to *PatientProfiler* as a "workflow". Text has been changed accordingly.

"Importantly, applying the analyte-wise (by row) and/or the sample-wise (by column) z-score." (page 3) this sentence sounds broken

The sentence was indeed broken and has been removed.

Typos found in the article

- "unbiasedly" should be "in an unbiased way" (several places)
- "from STEP 3" should be from "STEP 2" (Fig 1 caption)
- "rank" should be "ranked" (page 5)
- "jupyter" should be "python" (Fig 2 caption,)
- "phosphates" (page 6)

All the typos were fixed.

Reviewer #3

This study seeks to identify patient-specific models and breast cancer subtypes by combining multi-omic data with established signaling networks. The approach first infers protein activities from transcriptomic and phosphoproteomic data, then uses these predicted activities to define patient-specific signaling networks. These patient networks are clustered to identify seven breast cancer subtypes which are examined to understand the activation of cancer hallmark pathways and differences in patient prognosis. The study suggests that the Basal-like breast cancer subtype may be made up of two distinct signatures, with differences in prognosis.

POINT 27. The rationale for examining differential signaling and regulatory networks across cancer patients to define clinical subtypes and promote precision medicine is sound. However, a number of methodological aspects are unclear, several conclusions are inconsistent with the data presented, and some conclusions are overstated. The findings are largely consistent with known cancer biology and patient outcomes, such as poorer survival outcomes for patients with TNBC, but more work is needed to convincingly add to the current state of cancer knowledge. For example, Wang 2019 previously identified two basal-like subtypes, and the prognostic differences observed for other signatures in this study appear minimal. With significant refinements, this study could interest those working on tumor signaling and network approaches for multi-omic analysis.

We thank the reviewer for his/her/their appreciation, for carefully reading the manuscript and for the proposed improvements. In this new submission we have provided methodological details, gone across all the inconsistencies indicated by the reviewers. In particular, we agree that some of the biological conclusions, particularly regarding the Basal-like subtypes, have been overstated in the original version of the manuscript and as such we revised text to adopt a more measured tone and to avoid overstatements. We here would like to stress that our

intention was to demonstrate the power and robustness of the novel methodological framework we propose to derive novel insights from patient-specific multi-omic dataset. Breast cancer served primarily as a case study to illustrate the applicability of our network-based workflow for integrating multi-omic data.

In this line, we have extensively revised the manuscript to better reflect this focus. We have expanded the methodological sections of the manuscript by adding new paragraphs that clarify key methodological aspects of the approach and assessed its robustness (See also **Annex 1**). These changes aim to make the methodological contributions more prominent and to address the reviewer’s concern about the clarity and impact of our findings. Specifically, we have reduced the emphasis on biological novelty and included a more careful discussion of previously known subtypes, such as those identified in (Lehmann *et al*, 2011, 2016; Wang *et al*, 2019), both in the Introduction and in the Results sections. We now show that our Basal-like enriched communities (CL2 and CL4) are significantly associated with the well-characterized BL1 and BL2 subtypes, respectively (Benjamini-Hochberg correction, $p_{adj} = 1.22e-39$ and $1.38e-11$). This observation implies that the CL2 and CL4 causal networks generated may reflect the signaling events occurring in the well-characterized BL1 and BL2 subtypes. To validate this, we leveraged reverse phase protein array data (RPPA) derived from the very same TCGA patients. Specifically, we used this dataset to determine their coherence with the node activation state reported by *PatientProfiler* in the CL2 and CL4 networks. Although a limited number of nodes (12 out of 23 in CL2 and 5 out of 13 in the CL4 map) had been analysed in the RPPA dataset, we found that the activation status of key proteins measured in the RPPA dataset were consistent with the ones reported in the CL2 and CL4 networks. Specifically, as illustrated in the figure below (new **Figure 5C**), in line with our maps, CyclinE and MYC appear up-regulated in Signature2-enriched patients, whereas ERK2/MAPK1, PKC α /PRKCA and EGFR display an up-regulation in Signature4-classified patients. These observations are in line with results from (Lehmann *et al*, 2021). These new analyses have been incorporated as new text and new figures.

New Figure 5C. Difference in protein levels of CL2 and CL4 nodes extracted from the map in (A), as derived from the Reverse Phase Protein Arrays (RPPA) of TCGA patients (Koboldt *et*

al, 2012). Proteins up-regulated in Signature2- and Signature4-enriched TCGA patients are displayed in light blue and pink, respectively. On the right, the symbology indicates whether the protein level is coherent (green v) or incoherent (red x) with the activity in the map in (A). Only significant results (Benjamini-Hochberg correction, $p_{adj} < 0.05$) are displayed.

We hope that the analyses that have been carried out, along with text revisions will convince the reviewer that, altogether, our results provide evidence that the network-based stratification approach implemented via *PatientProfiler* can identify clinically and molecularly distinct subgroups of breast cancer patients. These subgroups not only correlate with established molecular classifications such as BL1 and BL2, but also provide possible mechanistic insights through pathway-specific alterations supported by proteomic data.

Other comments have been addressed as follows:

Specifically:

POINT 28. The analysis does not account for important covariates, including age or tumor mutational burden.

We tested if communities identified in CPTAC patients were associated with clinical and molecular covariates. Community 2 shows a significant positive association with the Tumor Mutational Burden ($p_{adj}=0.013$). No other covariates (Stage, Age, Ethnicity) were significantly associated with any communities. We included this analysis in the manuscript.

POINT 29. There is an over-reliance on the predictions of the 4 classical subtypes (Basal-I, Her2-I, LumA-I, and LumB-I) across the later analyses. This is confusing given the study premise that these 4 subtypes are insufficient for classifying patients, and are themselves a computational classification with potential for noise. Further, the patient stratification results are not provided for the 4 classical subtypes, leaving the benefit of additional subtypes unclear.

Classical subtypes continue to play a vital role in considering prognosis, treatment selection, and clinical trial design in breast cancer and also have proven to be successful in many patients (Nolan *et al*, 2023). Our scope is to demonstrate that *PatientProfiler* has the potential to fill the gap left by this classification and not to revolutionize the entire system. As such, with the present manuscript we first aim at demonstrating *PatientProfiler* can capture the differences across classical subtypes defined by the current classification and, next, to go beyond and provide more granular and orthogonal stratification.

To clarify our scope, we added a sentence in the introduction, and we have expanded the discussion.

POINT 30. The relationship of the results to the genomic profiles of each patient is unclear (is genomic information incorporated in PatientProfiler?), and limits the utility in a clinical setting. This genomic information could be utilized to measure and understand the coherence of the proposed subtypes.

Genomic data was indeed used as described in the method section. We have now revised the paragraph to ensure clarity (Pag 4). *Somatic mutation data was annotated as gain-of-function (GOF) or loss-of-function (LOF) using OncoKB API (<https://www.oncokb.org/>) (RRID:SCR_014782). We used the MafAnnotator.py OncoKB tool via a command line with Python 3 (RRID:SCR_024202) to annotate the impact of each mutation on protein function (1 for GOF and -1 for LOF) (Supplementary Table 2). We were able to obtain mutation-derived protein activities for 100 out of 122 patients. These were included in the list of patient-specific protein activities measured in the STEP2 of the workflow.*

POINT 31. The methodology underlying PatientProfiler lacks some details and could be streamlined to reduce repetition in the methods, supplemental methods, results, and discussion.

We sincerely apologize for the lack of details, methods and supplementary materials have now been extensively revised. To avoid repetitions, we have now provided a general and very detailed description of *PatientProfiler* in the extended methods, whereas its application to Breast Cancer data has been provided in the methods section of the main, added a novel **Supplementary Table 1** to clarify the parameters that were used. Additional points have been revised as detailed below.

POINT 32. (Step 1) What, if any, imputation was performed? Please clarify whether "Sample-wise z-scoring" is Z-scoring of each gene across all patients or all genes within a patient.

We apologize for the lack of clarity, because data are approximately normally distributed, we imputed missing values using the "norm" method of the "mice" package (Buuren & Groothuis-Oudshoorn, 2011). We have now specified this information in the methods paragraph. Sample-wise z-scoring refers to Z-scoring all genes within a patient (sample). To improve clarity, we have rephrased the sentence: "...computed a sample-wise z-score (i.e. based on the distribution of all the genes within a patient/sample)."

POINT 33. (Step 2) Please provide further details on the protein inference methods (e.g. VIPER) and rationale for parameter choices (e.g. correct_proteomics = FALSE for TFEA when the SignalingProfiler publication states that setting this to TRUE reduces RMSE)

We thank the reviewer for raising this point, as shown in the original *SignalingProfiler 2.0* publication, the correct_proteomics = TRUE parameter refers to a method that gives less

importance to phosphosites equally modulated in proteomics and phosphoproteomics. As such, its application make sense exclusively in phosphoproteomic studies, i.e. in the KSEA analysis of *SignalingProfiler/PatientProfiler*. In the original publication (Venafrà *et al*, 2024), it was applied when deriving the activity of Transcription factors from the PhosphoScore calculation (i.e. from the abundance of regulatory phosphosites in the phosphoproteomic dataset). In both *SignalingProfiler* and *PatientProfiler* there is just one unique function that handles the VIPER inference (for TFEA and KSEA), as such it's always necessary to specify the `correct_proteomics` parameter. It's important to stress, however, in the TFEA is recommended to set this parameter to `FALSE` (as visible in the associated code). To avoid confusion, we have removed the sentence.

POINT 34. (Step 4) The schematic implies that the bipartite community detection partitions all patients and proteins into a single community each. However, numerous proteins are present in multiple communities (E.g. MAPK1 in 2 & 4) - how similar are the different communities? How should one interpret the presence of both an up-regulated and down-regulated version of a protein in a community network (e.g. MAPK1 in community 6)? Could the confidence of the patient-community assignments be quantified? (Step 4)

As shown in **Figure 4A** (second panel), in the bipartite graph, patients (patient-nodes) are connected to the specific molecular interactions present in their regulatory networks (interaction-nodes). In other words, if the interaction $\text{MAP2K1} \rightarrow \text{MAPK1}$ (denoted as interaction j) is present in the regulatory model of patient i , the bipartite graph includes an edge between patient i and interaction j . This means that clustering is based on shared interactions and not on shared genes. As a result, the same gene may appear in multiple communities, but always within different regulatory contexts. Since this level of complexity may lead to ambiguous interpretations, we have revised text in the extended methods, further clarifying the bipartite projection, the terminology used and including the previous example in the manuscript.

In addition, in each patient-specific network, a gene can be either in an activated or inhibited state, depending on its regulatory context. To represent this distinction, each node is modeled as two separate entities: one corresponding to its activated form and the other to its inhibited form. Consequently, it is possible for the same gene to appear in both activated and inhibited forms within a single community, depending on co-occurring interactions across patients. We refer to these nodes/genes as ambiguous nodes. In the Figure below (new **Supplementary Figure 12**) we report the percentage of ambiguous nodes per community. As shown, this value tends to correlate with the network size (# Nodes). We here discuss that these ambiguous nodes probably reflect some noise in the network. However, their percentage remains low in most of the communities.

Breast Cancer detected communities, summary

Community		# Nodes	#Edges	#Patients	Ambiguous Nodes (%)	Derived Signature	# Genes in Signature
Community (CL1)	1	141	171	15	3.5	Signature 1	12
Community (CL2)	2	183	234	27	10.3	Signature 2	11
Community (CL3)	3	76	66	7	1.3	Signature 3	28
Community (CL4)	4	120	131	15	1.6	Signature 4	16
Community (CL5)	5	176	233	25	6.8	Signature 5	27
Community (CL6)	6	166	156	17	4.2	Signature 6	11
Community (CL7)	7	123	110	16	3.2	Signature 7	21

Supplementary Figure 12 Breast cancer communities, summary data. Information associated to each community: number of nodes, number of edges, number of patients, percentage of ambiguous nodes (i.e. nodes that appear as both active and inactive in the relative community), associated transcriptomic signatures, number of genes in the transcriptomic signatures.

Overall, we understand that a general description and discussion of the communities has been overlooked. To address this issue, we included a novel paragraph and complemented it with new Figures and analyses (Supplementary Figures 12-13).

Supplementary Figure 13 Similarity across Communities. The heatmap shows the overlap between detected communities (Jaccard's similarity). Each value represents the Jaccard index between pairs of communities, calculated from the set of genes assigned to each. The generally low similarity scores suggest a significant structural difference across communities.

POINT 35. What are the impacts (e.g. proportion of patient networks excluded) for each of the filtering steps applied to G^* ?

We agree that it is important to clarify which types of interactions are considered during the community detection step, how they directly influence the results, and the logic behind excluding others.

Our method includes a filtering step on interaction-nodes based on two-degree thresholds:

1. it removes rare or potentially noisy interaction-nodes linked to fewer than t_L patients;
2. it filters out frequent interaction-nodes connected to more than t_u patients, as these likely represent general (and probably well-known) biological processes, and are less informative for distinguishing patient communities.

There is no general optimal threshold choice, as the best parameters depend on the intrinsic characteristic of the cohort under analysis. For example, when aiming for a deeper stratification of patients (as in our case), it can be reasonable to set the upper threshold to

filter out interactions present in more than half of the patients. As for the lower threshold, we empirically excluded values equal to 1 as they correspond to interactions appearing in only one patient and therefore unlikely to support stratification. We then selected the smallest threshold that resulted in a modularity score above 0.3, a commonly accepted cut-off in the literature (Signorelli & Cutillo, 2022), above which clusters are generally considered to have a meaningful modular structure. This choice ensures that we extract densely connected communities, while still retaining sufficient information and avoiding aggressive filtering that might occur with higher values of t_u . In our case, $t_u=4$ resulted in a modularity value equal to 0.32, while $t_u=2$ and $t_u=3$ produced smaller values.

To better illustrate the effect of this filtering step, we included a pie chart (see new **Supplementary Figure 11B**) depicting the degree distribution of interaction-nodes, providing an overview of how the thresholds influence the number of interaction-nodes filtered out.

Supplementary Figure 11B. Distribution of interaction-nodes. The pie chart shows the percentage of interaction-nodes based on how many patients share them. Each slice corresponds to the proportion of interactions appearing in exactly 1, 2, 3, or 4 patients, as well as those shared by more than 4 patients (>4).

POINT 36. In Breast Cancer use case, it is stated that "mutation in TP53 seems to associate prevalently with LumA-I and Basal-I patients." However, SFig 1 shows only 4/25 LumA-I patients with TP53 mutations. Should this state "HER2-I", which would be consistent with Krug, 2020?

We apologize, mutation in TP53 is significantly enriched in Basal-I subtype (Fisher's exact test with Benjamini-Hochberg correction, $p_{adj} = 1.963e-08$) and significantly depleted in Luminal

subtypes. Subtype HER2-I shows significant association with the double mutation in TP53 and PIK3CA (Fisher's exact test with Benjamini-Hochberg correction, $p_{adj} = 0.02943$).

POINT 37. Further, it does not appear that all 122 patients are present in the analysis in SFig 1 - why is this?

We apologize if this point was not clear. To avoid misinterpretation, we revised the methods as follows: *somatic mutation data was annotated as gain-of-function (GOF) or loss-of-function (LOF) using OncoKB API (<https://www.oncokb.org/>). We were able to obtain mutation-derived protein activities for 100 out of 122 patients (Supplementary Figure 2, Supplementary Table 2).*

POINT 38. In Protein activity estimation, the statement "we observe that CDK2 and CDK5 are more active in Basal-I cell line as compared to cell lines representative of other subtypes" is inconsistent with the data presented.

We apologize if this point was not clear. In **Supplementary Figure 9** we observe that CDK2 activity is higher in Basal-I cells, this is shown by the combination of the phosphorylation at the activating residue pThr160 (even if not significant) and by the increased levels of the corresponding cyclin (significant). Although quantification of phosphorylation might appear greater in MDA-MB-361 cells (LumB-I), it is important to note that the bands are much weaker and the cyclin levels almost absent. Similarly, CDK5 is more active in a Basal-I cell line (MDA-MB-231), compared to cell lines representative of other subtypes as revealed by increased levels (significant) of phosphorylation of the **activating** pTyr15 site (Zukerberg *et al*, 2000).

We believe this confusion has derived from a different ordering of the samples in the Western blot images in respect to the quantification histograms. To improve clarity and ensure consistency, we have rearranged the columns of the quantification histograms shown in **Supplementary Figure 9**, accordingly.

Also, to facilitate the interpretation we revised the paragraph as reported in Pag. 12.

As an important side note, during the revision process and upon carefully checking the original Western blot images to address the reviewers' comments, we realized that the blot image labeled as CDK5 in Supplementary Figure 9 was inadvertently incorrect and corresponded to CDK2. This has been corrected with the appropriate CDK5 blot. Importantly, all quantifications were originally performed using the correct data, and we have re-verified their accuracy. The original blot images and associated raw data files (including image lab files and Excel quantifications) are available upon request should the reviewers wish to examine them.

POINT 39. Additionally, the hypothesis being tested in this experiment (and its novelty given prior work on CDKs in TNBC, such as 1 and 2) is unclear, and key methodological details and justifications are missing: How many replicates were performed? How was TPS quantified? What is the justification for using TPS rather than a more typical loading control, such as Actin?

Has the TPS image been cropped (One would expect detection of protein bands outside the range shown)?

We thank the reviewer for these thoughtful comments and the opportunity to clarify both the hypothesis and methodological details. For simplicity this answer has been broken down into different sections:

How many replicates were performed?

Regarding the methodological details, we added in the “**Supplementary Figure 9**” caption that Western Blot quantifications refer to three independent biological replicates and that TPS means “Total Protein Staining”.

Has the TPS image been cropped (One would expect detection of protein bands outside the range shown)?

We confirm that the TPS image was cropped. For clarity and transparency, full TPS gels have now been included as **Supplementary Figure 21**.

In this study, we focused on a specific molecular weight range corresponding to our proteins of interest. While additional bands outside this range may exist, uncropped full-length blots in **Supplementary Figure 22** are representative examples from one of three biological replicates per target, selected for clarity. These confirm the absence of significant off-target bands outside the displayed range.

How was TPS quantified? What is the justification for using TPS rather than a more typical loading control, such as Actin?

The total protein staining has been acquired using the “Ponceau S” function of the ChemiDoc MP Imaging System (Bio-Rad). Then, we estimated the total stain volumes per lane using the Image Lab software to get a loading control for quantification.

About the normalization control for western blot, several studies over the past decades have raised drawbacks regarding the reliability of commonly used housekeeping proteins, such as β -actin or GAPDH. First, the housekeeping genes vary in expression across different biological contexts, such as cell types and experimental conditions, thereby introducing bias unrelated to actual protein loading (Li & Shen, 2013; Ruan & Lai, 2007). Therefore, using these as calibrators between cell lines can introduce normalization errors due to biological variability, not true protein loading differences. Second, the high expression levels of the housekeeping proteins often result in signal intensities that exceed the linear dynamic range, compromising the detection and the quantification of low-abundant proteins on the same blot (Moritz, 2017).

For those reasons, we preferred using total protein staining detection for normalization of protein loading (Fosang & Colbran, 2015; Gilda & Gomes, 2013). This is also consistent with guidelines from LI-COR Biosciences (2021, technical guide), which explicitly and preferably recommend normalizing to total protein signal.

All relevant methodological details have been added to the method section.

The hypothesis being tested in this experiment (and its novelty given prior work on CDKs in TNBC, such as 1 and 2) is unclear.

As the reviewer suggested, several studies have already demonstrated critical roles for CDK1 and CDK2 in TNBC progression and therapy resistance (Tang *et al*, 2024; Guan *et al*, 2023; Jian *et al*, 2021; Nie *et al*, 2019; Liu *et al*, 2014)

Here, we show that *PatientProfiler* advances this understanding by quantitatively assessing CDK signalling activities across breast cancer subtypes in TNBC patients. This approach reveals differences in CDK2 and CDK5 activity, particularly highlighting the elevated CDK activity in basal-like TNBC. Specifically, the predicted CDK2 and CDK5 activities in patients with different breast cancer subtypes, as determined by *PatientProfiler*, were consistent with their activities observed in the analysed breast cancer cell lines. Notably, the basal-like subtype exhibited the highest CDK activity levels, suggesting that these cell lines may serve as suitable models for future focused experiments.

We hope these clarifications address the reviewer's concerns.

POINT 40. In Application: Communities' prioritization and transcriptome signatures extraction. Differences in phenoscores should be quantified statistically to support proposed biological differences between the subgroups.

As suggested, we included the statistics. **Figure 3C** and **Figure 4C** text has been modified accordingly.

POINT 41. Overrepresentation analysis uses the entire human proteome as background (SFig 9), but *PatientProfiler* only has access to 7,500 proteins (Discussion). Therefore, the enrichments could simply reflect known biases of the underlying network.

In **Supplementary Figure 16** (OLD SFig 9) we used the entire proteome as background because the enrichment analysis was carried out on the transcriptomic signatures, derived from the transcriptomic data (and not from SIGNOR). Since the transcriptomic data cover 23,131 transcripts, we hope that the reviewer will agree with us in assuming that the human proteome is a reasonable choice.

POINT 42. Why are only up-regulated transcripts considered when defining the transcriptomic signatures?

To the best of our knowledge, transcriptomic signatures are gene sets that reflect a biological condition, and they are, in most of the cases, used in a positive way: the expression of the signature is a positive marker of the condition. This is the reason why we only used up-

regulated transcripts. However, we acknowledge that this point should be clarified. We added a sentence in the main text and in the supplementary methods.

POINT 43. In Signature validation on an independent Breast Cancer cohort: please provide sample size information for the survival curves. In Figure 5B, there appear to be only a few patients represented by signature 4. Are these patients actually representative of the signature?

Sample size information for the survival curve has now been added to **Figure 5B**. We have now added the censored point to highlight the real size of the cohort which entirely reflects the signature.

#Comparison with other tools

POINT 44. In Comparison with other methods: The recasting of p-values as a 'similarity index' with an emphasis on comparison of non-significant values is confusing, especially as a non-significant p-value could result from an underpowered test rather than true similarity. One idea for a more robust approach would be to measure the similarity of genomic or transcriptomic profiles of patients assigned to each cluster.

We thank the reviewer for the suggestion. However, as outlined in the reply to reviewer 1 (please refer to POINT 1), we have decided to censor the quantitative comparison with other tools, and to limit the comparison to qualitative differences.

POINT 45. Also, it is stated that "PatientProfiler and tSNE+KMeans produce groups with clinically relevant survival differences", yet Figure 6B does not show any significant survival differences for tSNE+KMeans. How were the basal-specific signatures chosen? Were there exactly two basal-enriched subtypes from each method, or were the top two most enriched subtypes chosen? Was multiple-hypothesis correction performed?

We apologize for the confusion, in the submitted version of the manuscript we forgot to update old Figure 6. However, as mentioned in the previous point (please refer to POINT 1), we have decided to withdraw the quantitative comparison with other tools, and to limit the comparison to qualitative differences.

Minor points

POINT 46. "Workflow designed to unbiasedly integrate cancer-derived data" - what is meant by 'unbiasedly' in this context? One could argue that the choice of a signaling network and the filtering criteria imposed on the structure of G* introduce biases into the analysis.

By defining *PatientProfiler* as a "Workflow designed to unbiasedly integrate cancer-derived data..." we meant that the analysis is not guided by the user's research focus. However, we understand that the statement might be misinterpreted, as such, we replaced "unbiasedly" with "systematically".

POINT 47. Please specify all statistical tests performed by name, and specify the multi-hypothesis correction method.

All the statistical methods and multi-hypothesis correction methods have now been specified.

POINT 48. Please check the significance designations in SFig 3. E.g., for TOP2A it seems unlikely that the difference between LumA-I and LumB-I is more significant than the difference between Her2-I and LumB-I.

We apologize for the bug in the code, the problem has now been fixed and relative figures re-generated (now in **Supplementary Figure 6**).

POINT 49. Please check all figure references and panel labels for accuracy. E.g., A 'Figure 3D' is referenced but does not exist.

We thank the reviewer for spotting the inaccuracy. We double-checked all the labels.

POINT 50. Network diagrams lack a visual legend

We apologize, the graph legend has now been added to **Figure 3A, 5A, S10 and S17**.

POINT 51. In Figure 3A, alignment of the y-axes would enable better comparison between group.

The y-axes have now been aligned in the corresponding **Figure 3B**.

POINT 52. Typos in SFig 6: Panel C is not indicated, and the ladder for pTyr15-CDK5 is marked at 35kDa.

We thank the reviewer for spotting the inaccuracy. Typos were corrected in new **Supplementary Figure 9** (old SFig 6), panel C label was added.

POINT 53. Methods Patient Profiler Step 4 uses the term 'interaction' to refer to nodes in the bipartite graph, which leads to confusion as 'interaction' is typically used to refer to the edges. 'Interactor' or simply 'node' would be preferred.

As shown in **Figure 4A** (second panel), in the bipartite graph, patients (named patient-nodes) are connected to the specific molecular interactions present in their regulatory networks (named interaction-nodes). In other words, if the interaction MAP2K1 → MAPK1 (denoted as interaction j) is present in the regulatory model of patient i , the bipartite graph includes an edge between patient i and interaction j . We understand the possible confusion caused by our notation. To avoid ambiguity and improve clarity, we have introduced the example in the extended methods and revised the terminology throughout the manuscript, referring explicitly to interaction-nodes and patient-nodes in the bipartite graph. Please refer to POINT 34 for additional details.

POINT 54. Please explain the inconsistency in the number of nodes present in Figure 3A and the number of nodes per patient in Figure 3B (which specifies a minimum of ~100).

We apologise if we didn't make this point clear. As indicated in the method and in the result sections, for each patient in the cohort, we made available the full interactome generated (**Supplementary Figure 10**) and a subnetwork displaying the functional circuits impacting key cancer hallmarks (**Figure 3A**). To improve clarity and avoid confusion, we added the label “full interactome” in **Figure 3B**.

POINT 55. The discussion should clearly summarize and contextualize the mechanisms underlying differences across patients.

We are not sure to have understood the reviewer's point. With the intention of addressing this point, we have now revised the discussion to include identified molecular mechanisms associated with distinct patients/communities. The following paragraph has been added (Pag. 21):

Its application to our collection of 122 personalized models allowed us to detect seven communities of Breast cancer patients. Our community-based stratification, derived from mechanistic multi-omic modelling, partially overlaps with intrinsic subtypes (as shown by enrichment analysis) and captures orthogonal features. This discrepancy might be due to the limited capabilities of PatientProfiler (discussed below), but it could also mirror the inadequacy of the current classification in capturing the complexity and the heterogeneity of Breast Cancer patients and might reflect the added resolution provided by our integrative approach.

These communities pinpointed two subgroups of Basal-I patients, enriched by Signature2 and Signature4 and characterized by distinct prognostic values. Moreover, we showed that TCGA patients enriched by Signature2 and Signature4 significantly associate with well-known Basal-

I subgroups, namely BL1 and BL2 (Lehmann et al, 2011; Wang et al, 2019). By this analysis, we demonstrated that PatientProfiler enables the systematic characterisation of such subgroups while providing mechanistic insight into the signalling events regulated in each subgroup and in each individual, with possible impact for therapy formulation. Specifically, in Community 2 we observe sustained cell growth and cell-cycle progression mediated by the MYC-CDK4/CDK6 axis, while in Community 4, the activation of a NF-kappaB-mediated pro-inflammatory response and of pathways driving a metaplastic phenotype, with a resulting increase in aggressiveness and invasiveness. Importantly, these observations are in line with results from (Lehmann et al, 2021) and the activation of these axes was also partially confirmed by looking at the RPPA profiles of the patients.

POINT 56. The availability of the pipeline code and all generated networks is appreciated.

We sincerely thank the reviewer for the appreciation.

ANNEX 1, Validation of the network models

Reviewers highlighted the need to assess the sensitivity of *PatientProfiler* to noise and its robustness at the various stages of the workflow: the protein inference, the network construction, the community detection, and the signature generation levels.

We present here, for each level, our assessment.

Protein inference and network construction

The protein inference and the network construction steps are computed by the SignalingProfiler algorithm (Venafrà *et al*, 2024) implemented in *PatientProfiler*. In our recent publication, we extensively performed validation of the results and robustness to noise.

Community and signature detection

To test the reliability and the robustness of *PatientProfiler*, we implemented a randomization strategy and a cross-validation.

Randomization results

As suggested by the reviewer, we generated control signatures after randomly shuffling different proportions of gene expression data (25%, 50%, 75%, 100% of genes), using 50 rounds of gene-wise randomisation. This approach preserves gene-level distribution while disrupting sample-wise structure. As reported in new **Figure 6A**, the overlap between control and original signatures decreases steadily with higher randomization levels, suggesting the shuffling is reflected in control signatures, reaching near-zero at full randomization (**panel A**). Although the number of control signatures enriching patient groups in TCGA remains roughly compatible with known BRCA subtypes (groups > 4) up to 75% randomization (**panel B**), their prognostic value is significantly diminished (**panel C**). This is shown in **panel C**, where the distribution of adjusted p-values with Benjamini-Hochberg correction from Kaplan–Meier analyses shifts upward with increasing randomization. This degradation is mirrored in the comparison of subtype distributions between CPTAC and TCGA cohorts, which become increasingly dissimilar and largely unaligned at 100% shuffling (**panel D**). This confirms that the observed effects are not due to chance and that gene-wise randomization effectively disrupts the informative content of the original signatures.

Randomization results

Figure 6. Randomization analysis result. **A)** Percentage of genes from the original *PatientProfiler* run recovered in control signatures across across 50 randomization runs at increasing proportion of shuffled genes (5, 10, 15, 25, 50, 75, 100%) in CPTAC transcriptomics data. **B)** Number of control signatures enriching TCGA patients across proportions of randomized genes. **C)** Prognostic value of control signatures evaluated via Kaplan-Meier analysis. Each boxplot summarizes the adjusted p-values across runs for different shuffling percentages. The horizontal red dashed line marks the 0.05 significance threshold. Background shading indicates p-value ranges. **D)** Similarity between CPTAC and TCGA patient subtype distributions, defined by average cosine similarity across runs. The horizontal red dashed line marks the average cosine similarity of the original *PatientProfiler* run.

Cross Validation results

To assess the robustness of the community-specific signatures, we performed a cross-validation procedure based on random sampling. For each *PatientProfiler* community, we randomly split the CPTAC patients 50 times into training (70%) and test (30%) sets. For each

split, we derived community-specific signatures using the training set and used these signatures to enrich test patients.

We then evaluated: i) **signature robustness**, whether the signatures generated from the 70% training set were similar to those obtained from the full dataset; ii) **consistency in patient assignment**, whether the test patients were assigned to the same communities as in the original full run. To this aim, we calculated Precision, Recall, and F1 score, where:

- **True Positives (TP)** were patients assigned to the same community in both the original run and the cross-validation.
- **False Positives (FP)** were patients assigned to different communities;
- **False Negatives (FN)** were patients who did not enrich any signature in the cross-validation run but had been assigned to a community in the original analysis.

As shown in **Supplementary Figure 19**, communities exhibit varying levels of classification performance and signature stability. While some signatures, such as those derived for CL2, CL4, and CL7, are relatively stable across training subsets (**Supplementary Figure 19A**), others, particularly CL6, appear more sensitive to sampling variability. Signature stability tends to correlate with community size (**Supplementary Figure 19C**), whereas classification performance is more influenced by the average size of the gene signature (**Supplementary Figure 19D**). Despite these differences, most communities achieved F1 scores well above 0.5 (**Supplementary Figure 19B**), underscoring the overall robustness of the *PatientProfiler* classifier. In particular, CL2 and CL4 display strong performance and stable signatures, suggesting they represent well-defined and biologically cohesive patient subgroups, supported by their moderate-to-large patient sizes and balanced signature complexity.

Crossvalidation results

Supplementary Figure 19. Cross-validation results for community-specific classification performance. **A)** Percentage of signature genes recovered in each of 50 cross-validation runs per community. **B)** F1 score distributions showing classification performance across runs. **C)** Number of CPTAC patients assigned to each community in the full run. **D)** Average number of genes included in the signature for each community across the 50 cross-validation runs.

- Buuren S van & Groothuis-Oudshoorn K (2011) mice: Multivariate Imputation by Chained Equations in R. *J Stat Softw* 45: 1–67
- Fosang AJ & Colbran RJ (2015) Transparency Is the Key to Quality. *J Biol Chem* 290: 29692–29694
- Gilda JE & Gomes AV (2013) Stain-Free total protein staining is a superior loading control to β -actin for Western blots. *Anal Biochem* 440: 186–188
- Guan T, Li M, Song Y, Chen J, Tang J, Zhang C, Wen Y, Yang X, Huang L, Zhu Y, *et al* (2023) Phosphorylation of USP29 by CDK1 Governs TWIST1 Stability and Oncogenic Functions. *Adv Sci Weinh Baden-Wurt Ger* 10: e2205873
- Jian Y, Huang X, Fang L, Wang M, Liu Q, Xu H, Kong L, Chen X, Ouyang Y, Wang X, *et al* (2021) Actin-like protein 6A/MYC/CDK2 axis confers high proliferative activity in triple-negative breast cancer. *J Exp Clin Cancer Res CR* 40: 56
- Koboldt DC, Fulton RS, McLellan MD, Schmidt H, Kalicki-Veizer J, McMichael JF, Fulton LL, Dooling DJ, Ding L, Mardis ER, *et al* (2012) Comprehensive molecular portraits of human breast tumours. *Nature* 490: 61–70
- Krug K, Jaehnig EJ, Satpathy S, Blumenberg L, Karpova A, Anurag M, Miles G, Mertins P, Geffen Y, Tang LC, *et al* (2020) Proteogenomic Landscape of Breast Cancer Tumorigenesis and Targeted Therapy. *Cell* 183: 1436-1456.e31
- Lehmann BD, Bauer JA, Chen X, Sanders ME, Chakravarthy AB, Shyr Y & PiTENPOL JA (2011) Identification of human triple-negative breast cancer subtypes and preclinical models for selection of targeted therapies. *J Clin Invest* 121: 2750–2767
- Lehmann BD, Colaprico A, Silva TC, Chen J, An H, Ban Y, Huang H, Wang L, James JL, Balko JM, *et al* (2021) Multi-omics analysis identifies therapeutic vulnerabilities in triple-negative breast cancer subtypes. *Nat Commun* 12: 6276
- Lehmann BD, Jovanović B, Chen X, Estrada MV, Johnson KN, Shyr Y, Moses HL, Sanders ME & PiTENPOL JA (2016) Refinement of Triple-Negative Breast Cancer Molecular Subtypes: Implications for Neoadjuvant Chemotherapy Selection. *PLOS ONE* 11: e0157368
- Li R & Shen Y (2013) An old method facing a new challenge: re-visiting housekeeping proteins as internal reference control for neuroscience research. *Life Sci* 92: 747–751
- Liu Y, Zhu Y-H, Mao C-Q, Dou S, Shen S, Tan Z-B & Wang J (2014) Triple negative breast cancer therapy with CDK1 siRNA delivered by cationic lipid assisted PEG-PLA nanoparticles. *J Control Release Off J Control Release Soc* 192: 114–121
- Moritz CP (2017) Tubulin or Not Tubulin: Heading Toward Total Protein Staining as Loading Control in Western Blots. *Proteomics* 17
- Nie L, Wei Y, Zhang F, Hsu Y-H, Chan L-C, Xia W, Ke B, Zhu C, Deng R, Tang J, *et al* (2019) CDK2-mediated site-specific phosphorylation of EZH2 drives and maintains triple-negative breast cancer. *Nat Commun* 10: 5114
- Nolan E, Lindeman GJ & Visvader JE (2023) Deciphering breast cancer: from biology to the clinic. *Cell* 186: 1708–1728

- Ruan W & Lai M (2007) Actin, a reliable marker of internal control? *Clin Chim Acta Int J Clin Chem* 385: 1–5
- Signorelli M & Cutillo L (2022) On community structure validation in real networks. *Comput Stat* 37: 1165–1183
- Tang J, Li J, Lian J, Huang Y, Zhang Y, Lu Y, Zhong G, Wang Y, Zhang Z, Bai X, *et al* (2024) CDK2-activated TRIM32 phosphorylation and nuclear translocation promotes radioresistance in triple-negative breast cancer. *J Adv Res* 61: 239–251
- Venafra V, Sacco F & Perfetto L (2024) SignalingProfiler 2.0 a network-based approach to bridge multi-omics data to phenotypic hallmarks. *NPJ Syst Biol Appl* 10: 95
- Wang D-Y, Jiang Z, Ben-David Y, Woodgett JR & Zacksenhaus E (2019) Molecular stratification within triple-negative breast cancer subtypes. *Sci Rep* 9: 19107
- Zukerberg LR, Patrick GN, Nikolic M, Humbert S, Wu CL, Lanier LM, Gertler FB, Vidal M, Van Etten RA & Tsai LH (2000) Cables links Cdk5 and c-Abl and facilitates Cdk5 tyrosine phosphorylation, kinase upregulation, and neurite outgrowth. *Neuron* 26: 633–646

10th Sep 2025

Manuscript Number: MSB-2025-13041R-Q
Title: PatientProfiler: A network-based approach to personalized medicine
Author: Veronica Lombardi
Lorenzo Di Rocco
Eleonora Meo
Veronica Venafra
Elena Di Nisio
Valerio Perticaroli
Mihail Lorentz Nicolaeasa
Chiara Cencioni
Francesco Spallotta
Rodolfo Negri
Francesca Sacco
Livia Perfetto

Dear Livia,

Thank you for sending us your revised manuscript. We have now heard back from the three reviewers who agreed to evaluate your study. As you will see below, the reviewers agree that the study has been substantially improved, though they have raised a few relatively minor remaining issues.

Before we can accept your manuscript, we would ask you to address the following issues:

1. The remaining concerns of Referees # 1 and #2.

On a more editorial level:

2. Please submit the manuscript in .docx format.

3. Please upload each figure as an individual high-resolution file. Remove all figures from the manuscript file, but keep the figure legends in place below the References section.

4. Remove the "Author Contributions" section from the manuscript file.

5. The information on code availability should be incorporated into the Data Availability section, and the separate Code Availability heading should be removed.

6. The statement "The authors declare no potential conflicts of interest" should be removed from the title page and placed in a separate section under the heading "Disclosure Statement and Competing Interests."

7. Funding information: Please ensure that the funding information entered in the submission system is consistent with the information provided in the manuscript text. The following funders need to be in the system as separate entries (via More Funders option, Comments box should not be used) SEED Sapienza Grant; a PNRR fellowship Mission 4, Component 1, Action 4.1 "Research Doctorates", CUP:B53C23001770006 by the European UnionNextGenerationEU; PON-MUR fellowship (n. DOT13IEP1U-1); MUR - PRIN_2022(CUP:E53D23009910006), Sapienza University of Rome (Grant No.RG123188B4885C78); INdAM - GNCS Project 2024,CUP_E53C23001670001; Fondazione Umberto Veronesi.

8. Appendix: All the appendix figures should be bundled together with their legends in a single PDF file called *Appendix*, which should start with a Table of Content with page numbers that lists every figure and its page. Each legend should be below the corresponding Figure in the Appendix. Appendix figures should be referred to in the main text as: "Appendix Figure S1, Appendix Figure S2 " etc. See detailed instructions regarding expanded view here: <https://www.embopress.org/page/journal/17444292/authorguide#expandedview>.

9. Callouts: supplementary figures need to be called out as Appendix Figures S1-S22. Please add missing callouts for Appendix Figure S21 and S22.

10. The content currently under "Extended Methods" should be integrated into the main Methods section.

11. The four supplementary tables should be renamed to Dataset EV1-EV4. Please update the corresponding source file

names, include appropriate legends within each file, and revise all manuscript callouts accordingly.

13. Please address the following issues related to figure legends:

- Please note that the exact p values are not provided in the legends of figures 2F, 3C, 4C, 6C
- Please indicate the statistical test used for data analysis in the legend of figure 6C
- Please note that the box plots need to be defined in terms of minima, maxima, centre, bounds of box and whiskers, and percentile in the legends of figures 2F, 3C, 4C, 6A-D
- Please note that information related to n is missing in the legends of figures 2F, 3C, 4C, 6A-D

14. Please provide a "standfirst text" summarizing the study in one or two sentences (approximately 250 characters, including space), three to four "bullet points" highlighting the main findings and a "synopsis image" (550px width and 400-600 px height, PNG format) to highlight the paper on our homepage.

Here are a couple of examples:

<https://www.embopress.org/doi/10.15252/msb.20199356>

<https://www.embopress.org/doi/10.15252/msb.20209475>

<https://www.embopress.org/doi/10.15252/msb.209495>

15. When you resubmit your manuscript, please download our CHECKLIST (<https://www.embopress.org/pb-assets/embo-site/EMBO%20Press%20Author%20Checklist-1642513524327.xlsx>) and include the completed form in your submission.

Please note that the Author Checklist will be published alongside the paper as part of the transparent process (<https://www.embopress.org/page/journal/17444292/authorguide#transparentprocess>).

16. Please download and fill our Reagents and Tools Table template (.docx), which you can find in our author guidelines: <https://www.embopress.org/page/journal/17444292/authorguide#structuredmethods>.

17. The manuscript sections should be in the following order: Title page - Abstract & Keywords - Introduction - Results - Discussion - Methods - Data Availability - Acknowledgments - Disclosure Statement & Competing Interests - References - Figure Legends - (Main Tables with legends if applicable) - Expanded View Figure Legends.

Click on the link below to submit your revised paper.

Kind regards,
Jingyi

Jingyi Hou, PhD
Senior Editor
Molecular Systems Biology

*** PLEASE NOTE *** As part of the EMBO Press transparent editorial process initiative (see our Editorial at <https://dx.doi.org/10.1038/msb.2010.72>), Molecular Systems Biology will publish online a Review Process File to accompany accepted manuscripts. When preparing your letter of response, please be aware that in the event of acceptance, your cover letter/point-by-point document will be included as part of this File, which will be available to the scientific community. More information about this initiative is available in our Instructions to Authors. If you have any questions about this initiative, please contact the editorial office (msb@embo.org).

Reviewer #1:

I have seen the previous version of Lombardi et al and I found their responses to the many concerns raised properly addressed. The major issues with the presentation of this new tool have been resolved. The additional text, data and figures provide now the missing information. The terminology regarding PatientProfiler is now unified and the comparison with other tools is present and adequate. Carrying out only qualitative comparisons and no quantitative ones makes sense.

I only have a few minor comments to improve the clarify of the manuscript:

The title is too broad. While the manuscript has been correctly toned down throughout, the title also needs to be more specific to the workflow.

Similarly, the abstract is too general. It contains some very high level sentences and phrases, such as "causal interaction data to address how the genetic and molecular background of individual patients" ; "dysregulated signaling pathways" - these need to contain more specific information. Also, the end of the abstract is a bit incomplete, it should contain some outlook, additional applications closing.

There is a difficult to read sentence-paragraph in the Introduction: "As a matter of fact, the identification of signaling mechanisms that are deregulated at the patient resolution level and their employment in delivering more granular biomarkers represent an evolving field, limiting the design of patient-tailored therapeutic regimens"

Reviewer #2:

General remarks

The authors made a considerable effort to improve the article, especially regarding the clarity of the workflow and its parameterisation, and z-score normalisation. Some points require further clarification.

Major points

Calculation of molecular activity

In the revised version of the article the authors provide reasoning for their choice of sample-wise z-scores as the measurement of molecular activity. Supplementary Figure 4 features a heatmap of CPTAC transcriptomic data z-scores for known breast cancer biomarkers. Labels are added to match the four main categories, but the suggested patterns should be supported either by a dendrogram of the heatmap, a statistical test comparing the groups, or another method to quantify the inter-group differences. Currently, the reader is expected to infer these patterns by visual examination only.

Validation of the network models

In the newly added section "Robustness assessment" the authors evaluate the stability of their network models. It would be informative to evaluate how stable are network signatures when data are split before the entire workflow is executed, constructing network models for a randomly selected subset of patients. Currently it seems this split is done at the last step of the PatientProfiler pipeline, when the communities are already constructed based on the entire dataset. See Methods, "Each list of patients of the seven communities was randomly split into a training set (70%) and a test set (30%) across 50 independent iterations." Also, expressions "used to perform enrichment analysis on the test set" (Methods) and "enrich test patients" (Results) are unclear.

Regarding the "Randomization analysis", it focuses on the performance of already established signatures, using the same source dataset ("For each randomized dataset, PatientProfiler Step 5 was re-run to identify community-specific control signatures.") In this light, the statement "These findings confirm that the generated signatures reflect biologically meaningful signals rather than chance associations" (page 15) should be revised. Also there, statement "decreased reproducibility of patient subtypes between the CPTAC and TCGA datasets" is not supported by Figure 6D, where the cosine similarity across signatures remains high even for significant level of perturbation (75%).

Figures 6A and 6C sufficiently demonstrate the impact of randomisation on the performance of the signatures. Also, labelling the red line in Figure 6C will be helpful for the reader.

Minor points

Page 8, "The remaining mutations have only a mild effect on patient stratification" - is unclear

Page 10, "This is demonstrated by enhanced phosphorylation at the activating residue pThr160 (even if not significant) and increased levels of the associated cyclin protein." - what does "even if not significant" mean in this context?

Language

- Page 2, "critically, employed": "critically" is not needed in the sentence

- Page 2, "unbiasedly" should be "in an unbiased way"

Reviewer #3:

The authors have done a commendable job in addressing the first round of comments. Nothing further to add.

Reviewer #1:

I have seen the previous version of Lombardi et al and I found their responses to the many concerns raised properly addressed. The major issues with the presentation of this new tool have been resolved. The additional text, data and figures provide now the missing information. The terminology regarding PatientProfiler is now unified and the comparison with other tools is present and adequate. Carrying out only qualitative comparisons and no quantitative ones makes sense.

We sincerely thank the reviewer for critically re-reading our work and for the positive evaluation. We agree with the reviewer: given the significant change in tone of the manuscript a revision of the title and of the abstract is needed and has been carried out. We hope that the present form will satisfy the reviewer.

I only have a few minor comments to improve the clarify of the manuscript:

- The title is too broad. While the manuscript has been correctly toned down throughout, the title also needs to be more specific to the workflow.

We submitted a revised version of the title: "PatientProfiler: building patient-specific signaling models from proteogenomic data"

- Similarly, the abstract is too general. It contains some very high level sentences and phrases, such as "causal interaction data to address how the genetic and molecular background of individual patients" ; "dysregulated signaling pathways" - these need to contain more specific information. Also, the end of the abstract is a bit incomplete, it should contain some outlook, additional applications closing.

We revised the abstract.

- There is a difficult to read sentence-paragraph in the Introduction: "As a matter of fact, the identification of signaling mechanisms that are deregulated at the patient resolution level and their employment in delivering more granular biomarkers represent an evolving field, limiting the design of patient-tailored therapeutic regimens"

We revised this sentence as follows: "*As a matter of fact, the identification of patient-specific deregulated signaling mechanisms, and their translation into granular biomarkers, remains an evolving challenge that hampers the design of personalized therapeutic strategies (Eduati et al, 2020)*".

Reviewer #2:

General remarks

The authors made a considerable effort to improve the article, especially regarding the clarity of the workflow and its parameterisation, and z-score normalisation. Some points require further clarification.

We sincerely thank the reviewer his/her/their positive comments. Below we report a point-by-point rebuttal.

Major points

Calculation of molecular activity

In the revised version of the article the authors provide reasoning for their choice of sample-wise z-scores as the measurement of molecular activity. Supplementary Figure 4 features a heatmap of CPTAC transcriptomic data z-scores for known breast cancer biomarkers. Labels are added to match the four main categories, but the suggested patterns should be supported either by a dendrogram of the heatmap, a statistical test comparing the groups, or another method to quantify the inter-group differences. Currently, the reader is expected to infer these patterns by visual examination only.

We thank the reviewer for this suggestion. Our first intent was to make the heatmap in Supplementary Figure 4 (now Appendix Figure S4) comparable with the one from Nolan et al, 2023 to reassure the reviewer about our z-scoring approach. However, we understand that for an independent evaluation, a more unbiased presentation is needed. Clustering analysis has now been included in Appendix Figure S4.

Validation of the network models

In the newly added section "Robustness assessment" the authors evaluate the stability of their network models. It would be informative to evaluate how stable are network signatures when data are split before the entire workflow is executed, constructing network models for a

randomly selected subset of patients. Currently it seems this split is done at the last step of the PatientProfiler pipeline, when the communities are already constructed based on the entire dataset. See Methods, "Each list of patients of the seven communities was randomly split into a training set (70%) and a test set (30%) across 50 independent iterations." Also, expressions "used to perform enrichment analysis on the test set" (Methods) and "enrich test patients" (Results) are unclear.

We apologize for the lack of clarity. The generation of each patient network is totally independent from others' network generation and cannot be influenced by the size or by the composition of the cohort. As such, splitting data before the entire workflow is executed is not expected to change the results while having a great computational cost. Also, the main scope of this analysis is to determine the robustness of the community-derived signatures and their sensitivity to patient subsampling. Given this, we hope that the reviewer will agree with us that the present strategy is not incorrect. However, to make this point clearer we revised text. Also we revised indicated sentences.

Regarding the "Randomization analysis", it focuses on the performance of already established signatures, using the same source dataset ("For each randomized dataset, PatientProfiler Step 5 was re-run to identify community-specific control signatures.") In this light, the statement "These findings confirm that the generated signatures reflect biologically meaningful signals rather than chance associations" (page 15) should be revised. Also there, statement "decreased reproducibility of patient subtypes between the CPTAC and TCGA datasets" is not supported by Figure 6D, where the cosine similarity across signatures remains high even for significant level of perturbation (75%). Figures 6A and 6C sufficiently demonstrate the impact of randomisation on the performance of the signatures. Also, labelling the red line in Figure 6C will be helpful for the reader.

We agree with the reviewer, we removed unnecessary panels, we labelled the red line in Figure 6C (now Figure 6B) and revised text as indicated.

Minor points

Page 8, "The remaining mutations have only a mild effect on patient stratification" - is unclear

The sentence has been revised as follows: *"The remaining mutations did not show significant associations with molecular subtypes, suggesting they play only a limited role in patient stratification. This observation highlights the importance of signalling on tumorigenesis and cancer progression."*

Page 10, "This is demonstrated by enhanced phosphorylation at the activating residue pThr160 (even if not significant) and increased levels of the associated cyclin protein." - what does "even if not significant" mean in this context?

As shown in the bar diagram, the difference is not statistically significant. As such the general trend should be considered.

Language

- Page 2, "critically, employed": "critically" is not needed in the sentence

- Page 2, "unbiasedly" should be "in an unbiased way"

Language points have been addressed as suggested.

Reviewer #3:

The authors have done a commendable job in addressing the first round of comments.
Nothing further to add.

We sincerely thank the reviewer for critically re-reading our manuscript and for his/her/their positive feedback.

24th Sep 2025

Manuscript number: MSB-2025-13041RR

Title: PatientProfiler: A network-based approach to personalized medicine

Dear Livia,

Thank you again for sending us your revised manuscript. We are now satisfied with the modifications made and I am pleased to inform you that your paper has been accepted for publication.

Sincerely,
Jingyi

Jingyi Hou, PhD
Senior Editor
Molecular Systems Biology
